# VeriWeb: Verifiable Long-Chain Web Benchmark for Agentic Information-Seeking

## Abstract

Recent advances have showcased the extraordinary capabilities of Large Language Model (LLM) agents in tackling web-based information-seeking tasks. However, existing efforts mainly focus on single-fact retrieval and rely on outcome-only verification, thereby limiting their scalability in realistic knowledge-intensive scenarios that involve long-horizon web tasks requiring large-scale retrieval and synthesis of information from diverse sources. In this work, we introduce VeriWeb, a novel verifiable long-chain web benchmark designed to facilitate the evaluation and development of web agents within realistic web environments. Our benchmark emphasizes two critical dimensions: (1) *long-chain complexity*, encompassing both breadth- and depth-oriented search tasks to assess how effectively web agents ensure comprehensive information coverage and consistent context tracking in multi-hop reasoning; and (2) *subtask-level verifiability*, where tasks are decomposed into a sequence of interdependent verifiable subtasks. This structure enables diverse exploration strategies within each subtask, while ensuring that each subtask-level answer remains unchanged and verifiable. The benchmark consists of *302* tasks across five real-world domains, each with a complete trajectory demonstration, *annotated by human experts*. Extensive experiments on VeriWeb using various agents powered by different foundation models reveal significant performance gaps in handling long-horizon web tasks, highlighting the need for more powerful agentic information-seeking capabilities[1].

## 1 Introduction

Autonomous web agents have recently demonstrated remarkable capabilities in complex information-seeking tasks by following high-level instructions (Jin et al., 2025; Li et al., 2025c; Song et al., 2025a; Wu et al., 2025a;c; Gao et al., 2025), supporting a wide range of deep research systems (Google, 2025; OpenAI, 2025; xAI, 2025; MoonshotAI, 2025; Li et al., 2025e;e; Zheng et al., 2025). Recent breakthroughs in Large Language Models (LLMs) (Anil et al., 2023; Achiam et al., 2023; Guo et al., 2025; Yang et al., 2025; Zhang et al., 2025) have enabled promising prototypes of such agents, capable of complex search and browsing without relying on hard-coded automation or domain-specific scripting (Huang et al., 2025; Xi et al., 2025). However, developing such general-purpose web agents involves multiple complex processes, as it requires the ability to retrieve context-specific knowledge (Kwiatkowski et al., 2019; Joshi et al., 2017; Mallen et al., 2022), perform multi-hop reasoning across diverse sources (Press et al., 2024; Trivedi et al., 2022), and synthesize large-scale information (Du et al., 2025; Li et al., 2025e). This also poses a new challenge: how to obtain high-quality datasets that capture diverse, realistic information-seeking tasks to evaluate these agents effectively (Li et al., 2025b; Tao et al., 2025).

To address this challenge, various datasets and benchmarks have been released to advance the development of autonomous web agents (Mallen et al., 2022; Wu et al., 2025b; Wei et al., 2025; 2024; Mialon et al., 2024; Phan et al., 2025). Despite encouraging results, existing web benchmarks still exhibit two major limitations. *First*, most recent benchmarks focus on *single-fact retrieval* (Yang et al., 2018; Ho et al., 2020; Wei et al., 2025), where agents are tasked with retrieving an atomic fact, typically by performing shallow navigation and cross-page matching. For example, a task

---

[1] Anonymous code and data are available in the supplementary materials. Since the human demonstration data exceeds 100GB, only one sample is provided for reference due to the file size limit.

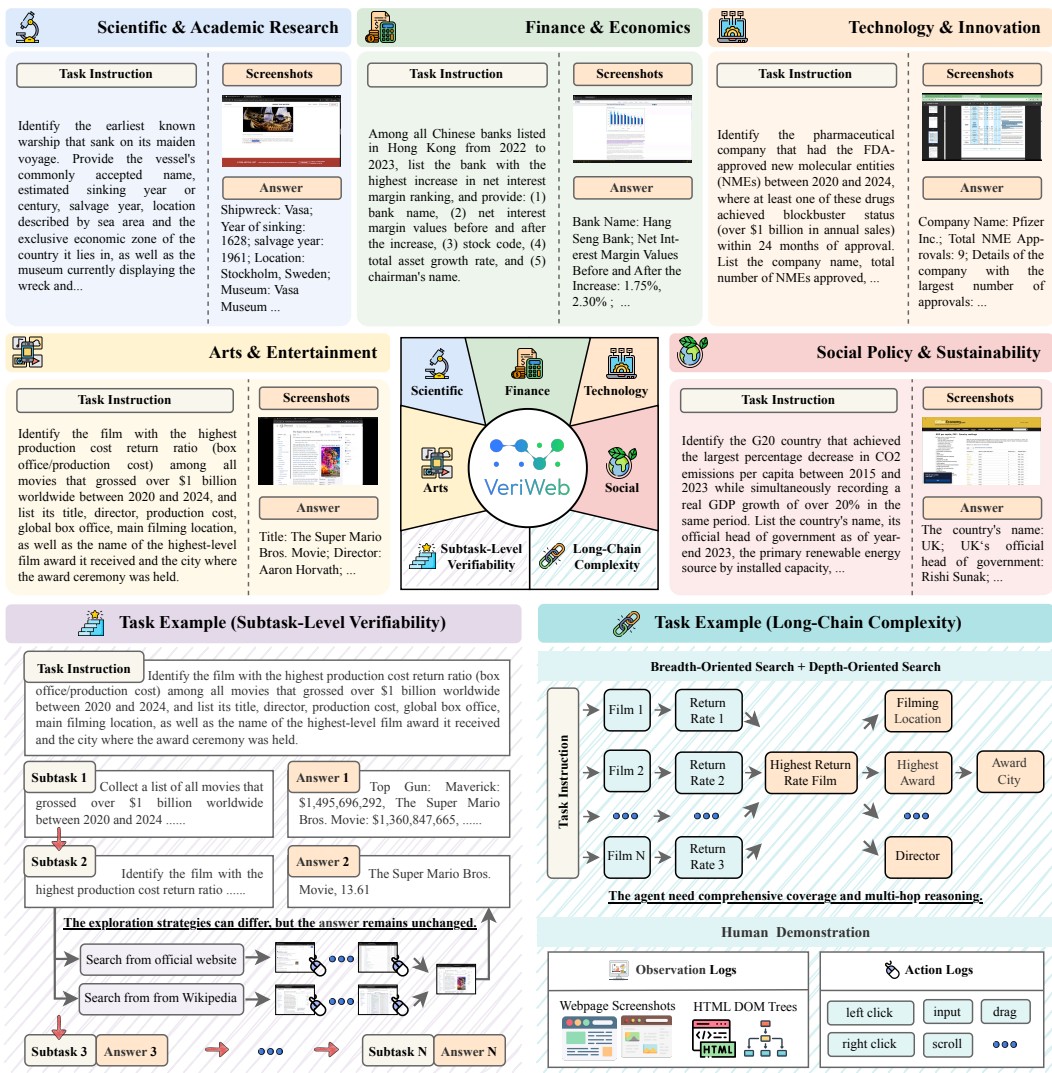

Figure 1: An overview of the VeriWeb benchmark across five domain-specific scenarios, which emphasizes (1) *long-chain complexity*, with tasks integrating both breadth- and depth-oriented search challenges, requiring comprehensive coverage and multi-hop reasoning. (2) *subtask-level verifiability*, where tasks are decomposed into interdependent subtasks with verifiable answers. Note that each task includes a complete human demonstration with detailed observation and action logs.

like "Which river that flows through Vienna also passes through Budapest?" can often be solved by simply navigating between two pages and extracting the answer "Danube". Such formulations rarely require large-scale retrieval and synthesis of information from diverse sources, both of which are essential for solving realistic information-seeking workflows that, most of the time, demand information-rich outputs rather than a single fact (Chatterji et al., 2025). **Second**, existing evaluation protocols typically rely on *outcome-only validation*, checking only whether the final result matches the ground-truth answer. This coarse-grained supervision fails to capture the quality of intermediate steps, especially when tasks involve multiple interdependent subtasks. In such cases, when agents fail, it is often unclear where or why the failure occurred, thereby making it difficult to support improvements to agent capability.

In this work, we introduce VeriWeb, a new verifiable long-chain benchmark tailored for the evaluation and development of web agents. VeriWeb comprises various web task trajectories across five domain-specific scenarios. All trajectories are carefully curated and annotated by human experts, ensuring long-chain complexity and subtask-level verifiability, as shown in Figure 1. (1) The *long-chain complexity* of VeriWeb features tasks that require agents to perform both breadth-oriented search (*e.g.*, listing all films with a box office over $1 billion) and depth-oriented search (*e.g.*, identi-

fying the film with the highest return rate → its highest award → the award city). To succeed, agents must engage in multi-hop reasoning, maintain coherent context across pages, and fuse fragmentary evidence into a well-supported synthesis. This design mirrors real-world information-seeking workflows, where relevant information is scattered across sources and must be reliably linked along a long reasoning chain. (2) The *subtask-level verifiability* of VeriWeb enables a fine-grained assessment of intermediate results at every subtask rather than solely at the final outcome. Notably, each subtask is designed to serve as a valid starting point, supporting agent evaluation at different task stages. A subtask consists of multiple steps involving realistic search and browsing operations. Instead of verifying each low-level action, the benchmark focuses on evaluating whether the goal of each subtask has been correctly achieved, providing a more informative supervision signal. This design supports open-ended interaction within each subtask, encouraging agents to explore diverse strategies to accomplish the subtask goal rather than adhering to a fixed action sequence. Our core contributions are summarized as follows:

- We curate a high-cost, human-annotated dataset of *302* verifiable long-chain task trajectories across five real-world domains, capturing both *long-chain complexity* and *subtask-level verifiability*. Each task is decomposed into interdependent subtasks with fixed, verifiable answers, emphasizing information-rich synthesis rather than single-fact retrieval.

- On top of this dataset, we design the VeriWeb benchmark, supporting multiple levels of evaluation, including task success rate, task completion rate, and action efficiency. This enables fine-grained analysis of agent capabilities across different stages of task execution and provides deeper insights into failure modes and information-seeking bottlenecks.

- Extensive experiments with a range of agents using state-of-the-art foundation models show consistently poor performance on long-horizon information-seeking tasks, underscoring current limitations of complex retrieval and synthesis in web agents.

## 2 RELATED WORKS

### 2.1 INFORMATION-SEEKING WEB BENCHMARKS

Existing web benchmarks broadly fall into two categories: interaction-centric and information-seeking tasks. The former focuses on UI-grounded action execution on the web, such as online shopping or emailing (Yao et al., 2022; Deng et al., 2023; Zhou et al., 2023). The latter targets search and browsing behaviors (Li et al., 2025b; Gao et al., 2025; Tao et al., 2025; Du et al., 2025)[2]. Early information-seeking benchmarks emphasize simple question answering over static corpora, typically solvable with single-hop queries (Kwiatkowski et al., 2019; Joshi et al., 2017; Mallen et al., 2022). Subsequent multi-hop benchmarks introduce compositional reasoning across multiple pages (Yang et al., 2018; Ho et al., 2020; Trivedi et al., 2022; Press et al., 2024; Wu et al., 2025b), but remain limited to Wikipedia-like closed environments. Recent efforts shift toward open-web scenarios (Wei et al., 2025; Zhou et al., 2025; Chen et al., 2025c; Wei et al., 2024; Chen et al., 2025a), though many still target single-fact retrieval, falling short of realistic tasks that demand information-rich, synthesized outputs. To bridge this gap, Du et al. (2025) benchmarks agents on report-level generation, aligning better with deep research workflows. However, such reports often include subjective or time-sensitive content, complicating direct ground-truth evaluation. Moreover, most benchmarks rely on outcome-only verification, neglecting challenges like error localization in long-horizon tasks. In contrast, VeriWeb is designed to reflect the complexity of real-world information-seeking tasks, supporting long-chain complexity and subtask-level verifiability.

### 2.2 INFORMATION-SEEKING WEB AGENTS

Information-seeking web agents have evolved from prompt-engineered browsing to reinforcement-learned "reason-with-search" behaviors (Sun et al., 2025; Jin et al., 2025; Li et al., 2025c; Song et al., 2025a;b; Chen et al., 2025b). These agents are now capable of deciding when and what to query, and of integrating retrieved evidence during multi-step reasoning (Wu et al., 2025a; Zheng et al., 2025; Li et al., 2025d; Geng et al., 2025). A growing line of work explores deep research agents (Li et al., 2025e;a; Qiao et al., 2025), which aim to perform end-to-end evidence gathering and report

---

[2]Note that this work focuses on information-seeking tasks, while interaction-centric tasks are out of scope.

synthesis, mirrored by production features like OpenAI Deep Research (OpenAI, 2025) and Gemini Deep Research (Google, 2025). Despite this momentum, our experiments show that current agents struggle with comprehensive coverage and multi-hop reasoning with consistent context tracking in complex information-seeking workflows, underscoring the need for benchmarks like VeriWeb that explicitly test long-horizon web tasks requiring large-scale retrieval and synthesis of information from diverse sources.

## 3 VERIWEB BENCHMARK

In this section, we present the task formulation, data collection procedure, and statistical analysis of the VeriWeb benchmark. VeriWeb is a carefully designed and human-curated benchmark featuring rigorous task formulation, expert annotation, and multi-stage review, yielding a challenging suite that targets real-world information-seeking tasks.

### 3.1 TASK FORMULATION

We formulate information-seeking tasks in VeriWeb as a Partially Observable Markov Decision Process (POMDP), defined by the tuple $\langle \mathcal{S}, \mathcal{O}, \mathcal{A}, P, O, R \rangle$, where $\mathcal{S}$ is the set of environment states, representing the full underlying web system. $\mathcal{O}$ is the observation space, and $O : \mathcal{S} \rightarrow \mathcal{O}$ is the observation function, which models the partial observations agents/humans receive from the environment. For web agents, the action space $\mathcal{A}$ consists of different tools (*e.g.*, search queries, webpage browsing), while the corresponding observations are the tool-call feedback. For human demonstrations, the action space $\mathcal{A}$ instead corresponds to user interactions such as mouse clicks or keyboard inputs, while the observations include webpage screenshots and the HTML DOM tree. $P : \mathcal{S} \times \mathcal{A} \times \mathcal{S} \rightarrow [0, 1]$ is the state transition function, modeling the dynamics of the web environment in response to actions. $R$ is the reward function, which is defined through verifiable answers.

For each information-seeking task in VeriWeb with an instruction $Q$, we collect a complete trajectory demonstration $\tau = (o_0, a_0, o_1, a_1, \ldots, o_T)$ from human annotators, where $T$ denotes the number of steps in the trajectory. To capture intermediate results and provide dense supervision, we decompose $\tau$ into a sequence of $K$ subtasks $\tau^{(1)}, \tau^{(2)}, \ldots, \tau^{(K)}$, such that $\tau = \tau^{(1)} \circ \tau^{(2)} \circ \cdots \circ \tau^{(K)}$, where $\circ$ denotes trajectory concatenation. The subtask $\tau^{(k)} = (o_{t_k}, a_{t_k}, \ldots, a_{t_{k+1}-1}, o_{t_{k+1}})$ corresponds to a contiguous segment of the full trajectory, where $t_k$ and $t_{k+1}$ denote the start and end timesteps. Each subtask $\tau^{(k)}$ is associated with a sub-instruction $Q^{(k)}$ and a subtask-level ground-truth answer $Y^{(k)}$. The task instruction $Q$ also has a task-level ground-truth answer $Y$.

### 3.2 DATA COLLECTION

**Data Source.** The VeriWeb dataset is constructed from a wide range of real-world web environments. We specifically focus on deep-research-like scenarios involving large-scale information retrieval and synthesis. Thus, we curate data from publicly accessible and authoritative sources as shown in Figure 2, including official websites of government agencies, academic institutions, online encyclopedias, financial databases, and news portals. These tasks cover five primary thematic domains: (1) scientific and academic research, (2) finance and economics, (3) technology and innovation, (4) arts and entertainment, and (5) social policy and sustainability. This categorization ensures diverse topical coverage and reflects realistic user intentions in complex information-seeking tasks.

**Task Instruction Construction.** To generate realistic and executable instructions, we develop a multi-stage pipeline combining human curation with language model generation, as shown in the left part of Figure 2. Initially, a small batch of seed instructions is manually selected for each topical domain. These seed instructions, representing high-level user intents, are input to a language model to generate a large number of candidate tasks. Human annotators then review these outputs, selecting only those that are grammatically clear, semantically meaningful, and practically feasible. Once a vetted pool of main tasks is established, the language model is prompted to perform subtask decomposition to obtain complete task instructions, including detailed sub-instructions of each subtask. This process is guided by seed instructions and strict formatting constraints. After generation, each batch of instructions undergoes automated filtering, followed by a second, stricter verification phase involving multiple passes of model-based validation. Only those tasks that pass

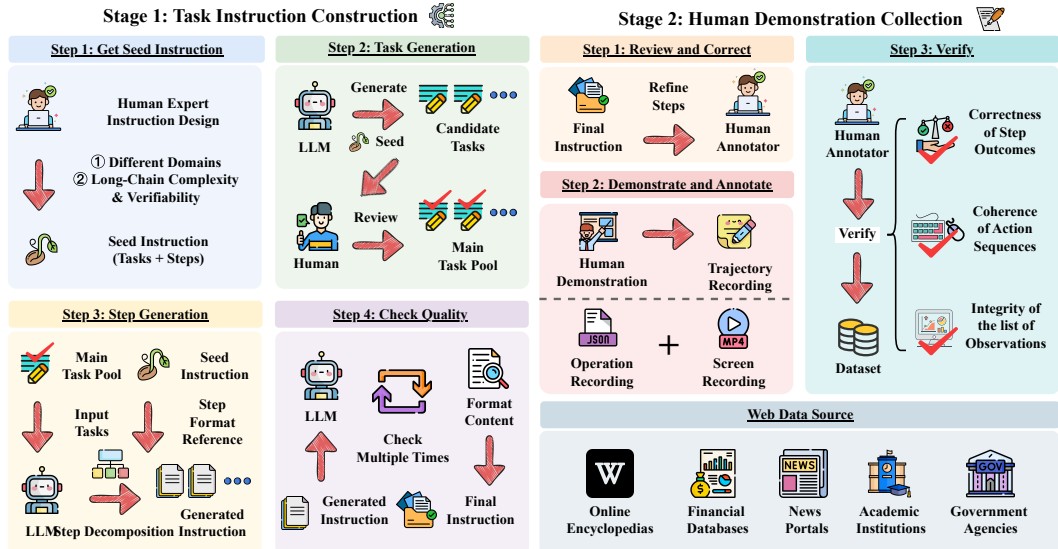

Figure 2: An overview of the proposed VeriWeb framework, consisting of two stages: task instruction construction and human demonstration collection. The framework combines LLM-based generation with human annotation to ensure realistic, high-quality web tasks and demonstrations.

all verification rounds are retained. This procedure enables efficient instruction generation while maintaining the factual correctness, diversity, and task feasibility necessary for web datasets. The detailed construction process can be found in Appendix A.

**Human Demonstration Collection.** Human annotators manually execute each task based on the given final instruction and record the complete trajectory demonstration, as shown in the right part of Figure 2. Before execution, human annotators refine the subtask sequence to ensure feasibility and smooth operation, allowing adjustments as needed during interaction. Demonstrations are recorded using screen capture tools, with detailed annotations including action logs, observation logs, and subtask-level goals. To ensure high-quality supervision and accurate benchmarking, all trajectory demonstrations undergo strict quality control. This includes both automatic checks and manual review to verify the correctness of subtask outcomes, coherence of action sequences, and integrity of observations. Only demonstrations that meet all criteria are retained. This guarantees that VeriWeb provides reliable and verifiable supervision for long-horizon web agents. The detailed collection process can be found in Appendix B.

## 3.3 DATA STATISTICS

To better understand the characteristics of Veri-Web, Figure 3 and Table 1 present statistical summaries of the collected human demonstrations for all tasks. The domain distribution of tasks in Figure 3a demonstrates that the dataset covers a wide range of domains, ensuring broad coverage and diversity across real-world tasks. Each task is decomposed into a sequence of subtasks, with each subtask associated with a verifiable answer. Figure 3c illustrates the distribution of subtasks per task, with an average of four subtasks per task. This subtask-level structure enables intermediate supervision, supporting more fine-grained evaluation and training. Veri-

Table 1: The overall data statistics of collected human demonstrations in VeriWeb.

| Statistic | Value |
|---|---|
| # Tasks | 302 |
| Avg. # Subtasks per Task | 4.3 |
| Avg. # Steps per Task | 272.5 |
| Avg. # Steps per Subtask | 63.5 |
| Avg. # Items of Task Answers | 5.7 |
| Max. # Items of Task Answers | 28 |
| Max. # Items of Subtask Answers | 94 |

Web further emphasizes long-chain complexity. As shown in Figure 3f, many tasks require executing hundreds of steps before completion. The large number of subtask answer items in Table 1 further highlights the need for large-scale retrieval and synthesis. Overall, these statistics demonstrate that VeriWeb provides both subtask-verifiable and long-chain tasks, offering a realistic and challenging benchmark for long-horizon reasoning and information-seeking in the web environment.

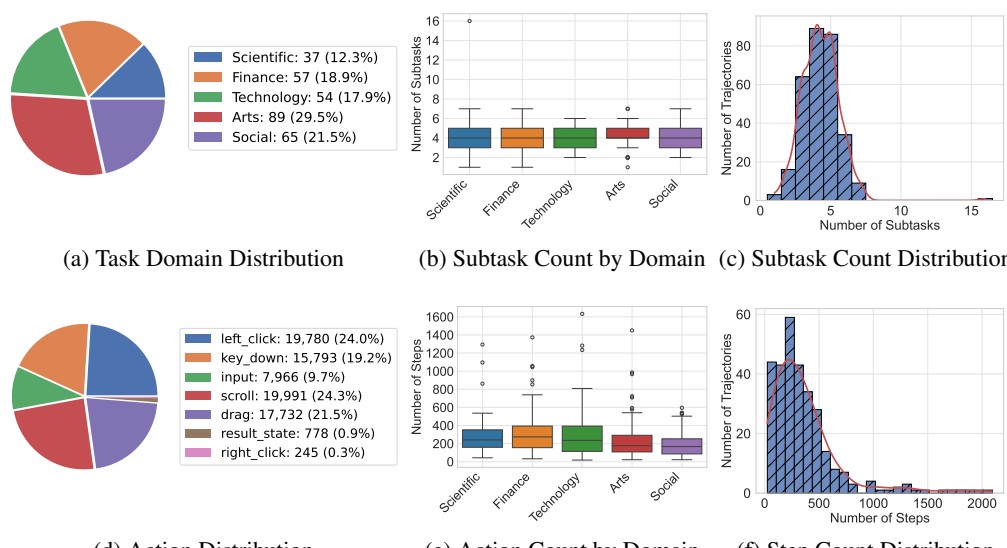

Figure 3: The detailed data statistics of collected human demonstrations in VeriWeb.

## 4 EXPERIMENTS

### 4.1 EXPERIMENTAL SETTINGS

**Baselines.** To demonstrate the effectiveness of VeriWeb, we compare four agent paradigms with different foundation models: (1) Deep research agents: closed-source systems with built-in search, including OpenAI Deep Research (OpenAI, 2025) and Gemini Deep Research (Google, 2025). (2) Search engine agents: models combined with an open-source search tool[3] via the model context protocol[4]. (3) Browser-use agents: models using the Browser-Use framework (Müller & Žunič, 2024). (4) Multi-agent systems: models using the Camel OWL framework (Hu et al., 2025). All used foundation models (Yang et al., 2025; Bai et al., 2025; Liu et al., 2024; Anil et al., 2023; Anthropic, 2024; Hurst et al., 2024) are shown in Table 2. Note that except for the closed-source deep research agents, search engine agents retrieve text-only context without page interaction, while browser-use and OWL agents operate on web elements and handle both text and visual input.

**Evaluation Metrics.** We evaluate agent performance using three metrics: (1) The *task Success Rate (SR)* measures whether the agent completes the overall task. (2) The *task Completion Rate (CR)* measures the extent to which the agent achieves the overall task goal. Since our tasks often involve multiple subtasks, CR estimates the completion level by calculating the proportion of correct items in the output. (3) The *Action Count (AC)* quantifies the planning effectiveness of agents by measuring the number of steps required to arrive at the final answer. Note that AE is not directly comparable across different agent paradigms and humans due to their inherently distinct action spaces. For both the SR and the CR, we use the LLM-as-a-Judge score (Gu et al., 2024) based on OpenAI-o3 to evaluate the correctness of the final answers. Detailed prompts are provided in Appendix C. Due to the high cost of API calls, each experiment is conducted once to obtain the final evaluation results.

**Evaluation Setting.** Our task involves long-chain complexity, encompassing both breadth- and depth-oriented search. We observe that the first subtask emphasizes breadth search, while subsequent ones focus on depth search. Therefore, besides evaluating the overall task, we also report results under two specific settings: (1) Breadth-oriented search, which evaluates only the first subtask. (2) Depth-oriented search, which evaluates the overall task given the result of the first subtask.

---

[3]https://github.com/searxng/searxng-docker
[4]https://modelcontextprotocol.io/

Table 2: Comparison of different agents on VeriWeb across five domains. Note that the browser-use agent results in Table 2–4 are updated because we identify some tasks with empty outputs (mostly due to unstable APIs) and re-run them during the rebuttal period.

| Model | Scientific | | Finance | | Technology | | Arts | | Social | | Overall | |
|---|---|---|---|---|---|---|---|---|---|---|---|---|
| | SR (%) | CR (%) | SR (%) | CR (%) | SR (%) | CR (%) | SR (%) | CR (%) | SR (%) | CR (%) | SR (%) | CR (%) |
| *Deep Research Agents* | | | | | | | | | | | | |
| Gemini-2.5-Flash | 0.0 | 19.7 | 1.8 | 19.6 | 0.0 | 28.3 | 19.1 | 48.7 | 4.6 | 26.2 | 6.9 | 31.2 |
| Gemini-2.5-Pro | 5.4 | 28.1 | 0.0 | 15.6 | 9.3 | 35.6 | 21.3 | 52.9 | 7.7 | 32.2 | 10.3 | 35.3 |
| OpenAI-o4-mini | 2.7 | 18.1 | 0.0 | 12.6 | 3.7 | 18.0 | 14.6 | 41.3 | 6.2 | 27.2 | 6.6 | 25.9 |
| OpenAI-o3 | 5.4 | 24.9 | 0.0 | 23.5 | 3.7 | 30.0 | 15.7 | 51.7 | 12.3 | 37.8 | 8.6 | 36.2 |
| *Search Engine Agents* | | | | | | | | | | | | |
| OpenAI-o3 | 0.0 | 17.6 | 0.0 | 18.2 | 7.4 | 28.0 | 13.5 | 37.6 | 3.1 | 20.6 | 6.0 | 26.1 |
| GPT-5 | 2.7 | 21.1 | 1.8 | 16.3 | 5.6 | 30.4 | 24.7 | 51.8 | 4.6 | 28.3 | 9.9 | 32.5 |
| *Browser-Use Agents* | | | | | | | | | | | | |
| Qwen-VL-Max | 0.0 | 3.8 | 1.8 | 2.6 | 0.0 | 3.3 | 1.1 | 11.0 | 0.0 | 11.4 | 0.7 | 7.3 |
| DeepSeek-V3.1 | 0.0 | 7.0 | 0.0 | 6.1 | 1.9 | 10.2 | 4.5 | 25.2 | 4.6 | 12.3 | 2.6 | 13.9 |
| Gemini-2.5-Flash | 0.0 | 4.6 | 0.0 | 7.5 | 0.0 | 10.2 | 7.9 | 27.9 | 0.0 | 11.1 | 2.3 | 14.4 |
| Gemini-2.5-Pro | 2.7 | 11.1 | 0.0 | 9.8 | 3.7 | 16.9 | 18.0 | 40.4 | 1.5 | 24.0 | 6.6 | 23.3 |
| Claude-3.7-Sonnet | 0.0 | 13.2 | 1.8 | 12.3 | 3.7 | 22.0 | 15.7 | 45.4 | 1.5 | 17.5 | 6.0 | 25.0 |
| Claude-4.0-Sonnet | 0.0 | 8.9 | 0.0 | 2.5 | 0.0 | 7.8 | 10.1 | 24.5 | 1.5 | 9.5 | 3.3 | 12.2 |
| OpenAI-o3 | 2.7 | 21.4 | 1.8 | 15.3 | 3.7 | 23.9 | 16.9 | 40.9 | 1.5 | 22.6 | 6.6 | 26.7 |
| GPT-4o | 0.0 | 7.3 | 0.0 | 1.2 | 0.0 | 10.2 | 4.5 | 22.7 | 0.0 | 10.6 | 1.3 | 11.9 |
| GPT-4.1 | 0.0 | 15.1 | 1.8 | 8.4 | 0.0 | 13.1 | 10.1 | 37.4 | 1.5 | 21.1 | 3.6 | 21.4 |
| GPT-5 | 0.0 | 14.1 | 1.8 | 4.9 | 1.9 | 9.6 | 7.9 | 21.8 | 0.0 | 17.5 | 3.0 | 14.6 |
| *Multi-Agent Systems* | | | | | | | | | | | | |
| Qwen3-235B | 0.0 | 8.4 | 0.0 | 3.3 | 0.0 | 6.7 | 2.2 | 19.0 | 0.0 | 12.3 | 0.6 | 11.1 |
| DeepSeek-V3.1 | 0.0 | 7.3 | 0.0 | 1.4 | 0.0 | 3.7 | 5.6 | 12.6 | 3.1 | 11.7 | 2.3 | 8.0 |
| OpenAI-o3 | 2.7 | 21.6 | 1.8 | 12.6 | 3.7 | 20.0 | 16.9 | 44.5 | 6.2 | 25.2 | 7.6 | 27.2 |
| GPT-5 | 2.7 | 14.6 | 7.0 | 9.5 | 0.0 | 5.6 | 5.6 | 23.3 | 9.2 | 18.3 | 5.3 | 15.4 |

## 4.2 MAIN RESULTS

Table 2 reports the agent performance on VeriWeb across five domains, measuring both task success rate and completion rate. Overall, the results highlight the difficulty of VeriWeb: no single configuration achieves more than a 15% success rate or a 40% completion rate. This underscores the challenging nature of VeriWeb, which involves large-scale retrieval, multi-hop reasoning, and complex information synthesis. We analyze the results from three perspectives: foundation model capability, agent paradigm, and domain-specific behavior.

**Foundation Model Comparison.** Performance differences across foundation models are substantial. Within deep research agents, OpenAI-o3 and Gemini-2.5-Pro stand out, demonstrating relatively strong reasoning and task generalization, while OpenAI-o4-mini lags behind. For search engine agents, GPT-5 can achieve better performance than OpenAI-o3. In browser-use agents, Qwen-VL-Max demonstrates limited effectiveness. Among multi-agent systems, OpenAI-o3 again yields the best results, while Qwen3-235B and DeepSeek-V3.1 struggle significantly. Interestingly, GPT-5 shows only moderate gains, pointing to open challenges in translating foundation model strength into effective agentic performance.

**Impact of Agent Paradigms.** The paradigm adopted by the agent strongly influences performance. Although the search engine agent relies on a simple search tool, its performance is broadly comparable to the other agents on VeriWeb, suggesting that effective use of search remains the dominant capability. Adding more complex tools, such as browser control or multi-agent coordination, does not necessarily yield further gains, and in some cases, the additional decision-making overhead may even hinder planning. Deep research agents demonstrate the highest overall CR, benefiting from stronger retrieval and summarization pipelines, with models like Gemini-2.5-Pro and OpenAI-o3 maintaining relatively balanced performance. Multi-agent systems show potential, as collaborative reasoning boosts robustness in certain domains.

**Performance Across Domains.** We also examine performance across the five domains to understand how content type affects agent effectiveness. Tasks in *arts and entertainment* generally saw the highest success and completion rates, likely due to the relatively clear and concrete nature of the information required. In contrast, domains like *finance and economics* and *social policy and sustainability* were more challenging, often requiring the agent to process fragmented, abstract information

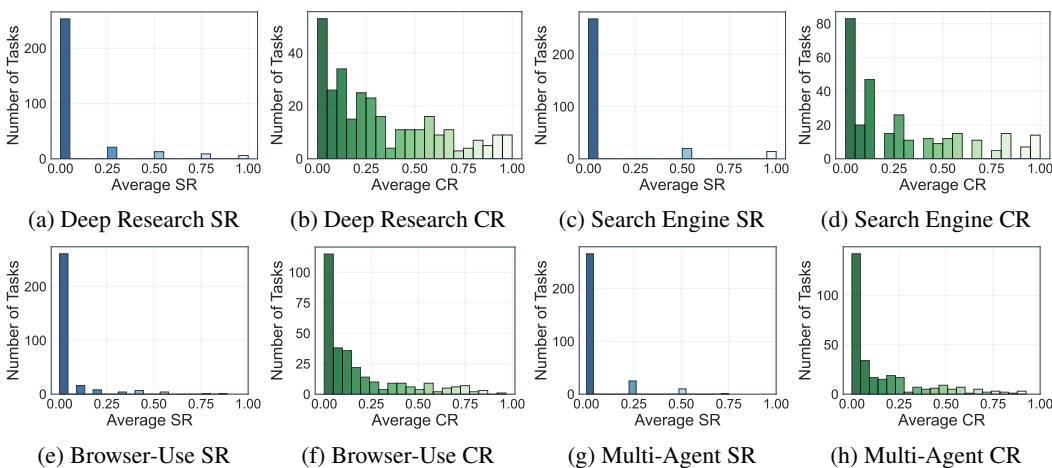

Figure 4: Distribution of task success rate (SR) and completion rate (CR) on VeriWeb.

from less standardized content. Most models performed poorly in these areas. The *scientific and academic research* and *technology and innovation* domains presented intermediate difficulty, involving complex technical descriptions or multi-attribute reasoning. These patterns indicate that the complexity of information presentation plays a crucial role in information-seeking tasks.

Table 3 compares model performance under breadth-oriented and depth-oriented settings. We observe an interesting phenomenon that breadth-oriented tasks often obtain a clearly higher task success rate, while the task completion rate is of similar magnitude in both settings. This reflects how errors accumulate and how the metrics are defined. In the breadth-oriented setting, the pipeline is relatively shallow and subtasks are weakly coupled, so local retrieval er-

Table 3: Comparison of different settings on VeriWeb.

| Model | Breadth-Oriented | | Depth-Oriented | | Overall | |
|---|---|---|---|---|---|---|
| | SR (%) | CR (%) | SR (%) | CR (%) | SR (%) | CR (%) |
| *Deep Research Agents* | | | | | | |
| Gemini-2.5-Flash | 17.5 | 43.1 | 7.6 | 36.1 | 6.9 | 31.2 |
| OpenAI-o4-mini | 17.2 | 37.4 | 8.3 | 43.4 | 6.6 | 25.9 |
| *Search Engine Agents* | | | | | | |
| OpenAI-o3 | 18.2 | 38.7 | 10.6 | 44.3 | 6.0 | 26.1 |
| GPT-5 | 23.2 | 38.6 | 15.6 | 45.5 | 9.9 | 32.5 |
| *Browser-Use Agents* | | | | | | |
| Claude-3.7-Sonnet | 16.6 | 37.8 | 10.6 | 40.1 | 6.0 | 25.0 |
| Claude-4.0-Sonnet | 11.3 | 20.8 | 6.0 | 18.6 | 3.3 | 12.2 |
| GPT-4.1 | 9.6 | 31.5 | 11.9 | 43.4 | 3.6 | 21.4 |
| GPT-5 | 12.9 | 19.7 | 4.6 | 17.9 | 3.0 | 14.6 |

rors rarely invalidate the entire instance, leading to a higher chance of fully correct solutions and thus higher SR. In the depth-oriented setting, later decisions depend on earlier ones, so any mistake in intermediate retrieval, reasoning, or synthesis can cause the whole instance to fail, which sharply reduces SR. However, CR is computed at the item level and still rewards trajectories that retrieve many correct pieces of evidence or partially complete the task, so depth-oriented CR remains comparable to breadth-oriented CR. Overall, current agents can perform broad retrieval reasonably well, but are fragile when required to maintain correctness over long, interdependent reasoning chains.

## 4.3 ANALYSIS

**Analysis of Task Difficulty.** To better understand the intrinsic difficulty of tasks in VeriWeb, we conduct a fine-grained statistical analysis of SR and CR distributions across all tasks, comparing results from different agent paradigms. For each task, the success rate is the average over all models within that agent paradigm. The distribution curves in Figure 4 reveal that for both agent paradigms, the majority of tasks yield low SR and CR values, with a long tail of near-zero success, underscoring the challenge of VeriWeb's long-chain requirements. To systematically categorize task difficulty, we define five levels based on the average SR and CR across all models and agents: (1) Level 1 includes tasks with SR above 0%, indicating they are relatively tractable for current agents. (2) Level 2 includes tasks with zero SR but CR above 15%. (3) Level 3 includes tasks with zero SR but CR between 5% and 15%. (4) Level

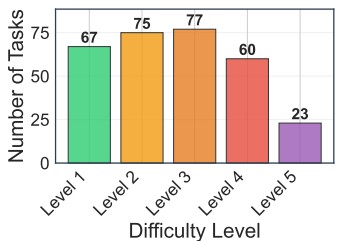

Figure 5: Task Difficulty Level.

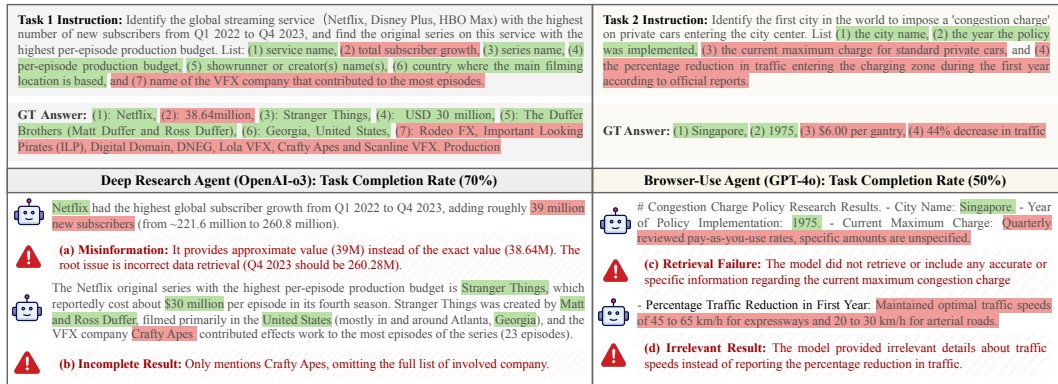

Figure 6: Case studies of agent performance on two information-seeking tasks in VeriWeb.

4 includes tasks with zero SR but CR between 0% and 5%. (5) Level 5 includes tasks where both SR and CR are zero, indicating no model was able to make progress. The results in Figure 5 show that most tasks fall into Levels 2-5 with zero SR, indicating high-complexity tasks. Only a minority of tasks fall into Level 1, suggesting that relatively few tasks are easily solvable. This categorization provides a practical framework for benchmarking future agent progress.

**Analysis of Action Efficiency.** The analysis of action efficiency reveals notable differences in the searching strategies of browser-use agents powered by different foundation models. As shown in Table 4, models such as Gemini-2.5-Flash generally require more actions for information-seeking, suggesting a more exploratory style. In contrast, models like OpenAI-o3 tend to accomplish tasks with fewer steps, indicating a more direct strategy. However, lower action counts do not necessarily correlate with higher success rates. Conversely, higher action counts sometimes reflect more thorough exploration, which proves advantageous in tasks involving complex or ambiguous objectives.

Table 4: Comparison of action count for browser-use agents.

| Model | Average AC |
|---|---|
| Qwen-VL-Max | 40.6 |
| DeepSeek-V3.1 | 44.3 |
| Gemini-2.5-Flash | 48.7 |
| Gemini-2.5-Pro | 46.6 |
| Claude-3.7-Sonnet | 41.7 |
| Claude-4.0-Sonnet | 25.7 |
| OpenAI-o3 | 18.9 |
| GPT-4o | 56.5 |
| GPT-4.1 | 23.6 |
| GPT-5 | 39.4 |

## 4.4 CASE STUDIES

To better understand agent behaviors and limitations in information-seeking tasks, we present two representative cases from VeriWeb in Figure 6. These examples illustrate retrieval fidelity, multi-hop reasoning quality, and four typical failure modes: (a) misinformation, (b) incomplete result, (c) retrieval failure, and (d) irrelevant result. For Task 1, the agent performed relatively well, correctly identifying the service, series, and most metadata. However, it introduced *misinformation* by reporting an approximate subscriber growth figure instead of the exact value, and gave an *incomplete result* by listing only

Table 5: Comparison of different failure modes on VeriWeb.

| Model | Misinformation | Incomplete Result | Retrieval Failure | Irrelevant Result |
|---|---|---|---|---|
| *Deep Research Agents* | | | | |
| Gemini-2.5-Flash | 44% (254) | 34% (197) | 14% (82) | 8% (42) |
| Gemini-2.5-Pro | 44% (213) | 36% (172) | 11% (54) | 9% (38) |
| OpenAI-o4-mini | 50% (249) | 36% (178) | 9% (46) | 5% (20) |
| OpenAI-o3 | 63% (248) | 30% (118) | 3% (12) | 4% (11) |
| *Search Engine Agents* | | | | |
| OpenAI-o3 | 62% (226) | 31% (113) | 4% (16) | 3% (9) |
| GPT-5 | 46% (172) | 42% (158) | 6% (23) | 6% (16) |
| *Browser Use Agents* | | | | |
| Qwen-VL-Max | 12% (51) | 67% (274) | 14% (58) | 7% (24) |
| DeepSeek-V3.1 | 21% (100) | 55% (262) | 17% (81) | 7% (27) |
| Gemini-2.5-Flash | 25% (125) | 53% (258) | 12% (58) | 10% (40) |
| Gemini-2.5-Pro | 34% (149) | 49% (214) | 12% (53) | 5% (19) |
| Claude-3.7-Sonnet | 34% (189) | 38% (212) | 19% (105) | 9% (40) |
| Claude-4.0-Sonnet | 19% (85) | 58% (253) | 16% (73) | 7% (24) |
| OpenAI-o3 | 50% (183) | 39% (144) | 5% (20) | 6% (16) |
| GPT-4o | 22% (125) | 47% (269) | 24% (140) | 7% (34) |
| GPT-4.1 | 33% (216) | 37% (246) | 24% (158) | 6% (34) |
| GPT-5 | 17% (81) | 55% (254) | 20% (94) | 8% (28) |
| *Multi Agent Systems* | | | | |
| Qwen3-235B | 23% (97) | 56% (239) | 12% (53) | 9% (31) |
| DeepSeek-V3.1 | 12% (55) | 61% (270) | 17% (76) | 10% (36) |
| OpenAI-o3 | 42% (168) | 41% (163) | 8% (32) | 9% (32) |
| GPT-5 | 19% (79) | 60% (243) | 13% (56) | 8% (25) |

one VFX company. For Task 2, the agent identified the correct city and year but failed in two key areas. It suffered a *retrieval failure* by not providing a specific congestion charge, and produced an

*irrelevant result* by reporting traffic speeds rather than the required percentage reduction. Beyond individual examples, our experiments also reveal several systemic limitations. First, the agents often demonstrate shallow search behavior, invoking tools only a few times and stopping early. This limits their ability to perform comprehensive, multi-hop retrieval. Second, the agents often formulate web queries using full sentences rather than concise keywords, leading to suboptimal results and reduced accuracy.

We further conduct a quantitative breakdown of different failure modes using GPT-5 as a judge. The detailed evaluation prompt is provided in Appendix C. Note that a single response may belong to several failure modes at the same time. The results in Table 5 show that misinformation and incomplete results are the two dominant failure modes across all agent paradigms, while retrieval failures and irrelevant results occur less frequently but are still non-negligible. Deep research and search engine agents tend to fail primarily through misinformation, suggesting that they often retrieve relevant evidence but hallucinate or approximate key numerical or factual details. In contrast, the browser-use agents and multi-agent systems are more frequently dominated by incomplete results, indicating that these agents are better at locating relevant sources but struggle to aggregate and exhaustively cover all required items in long-chain tasks. Across all settings, retrieval failures rarely account for the majority of errors, highlighting that the central bottleneck lies in accurate synthesis and coverage rather than merely finding at least one relevant document.

## 4.5 LLM-AS-A-JUDGE AGREEMENT

We further compare human and LLM judgment to validate the stability and reliability of our evaluation protocol. Since the deep research agent using OpenAI-o3 performs well, we use its responses for this analysis. In our experiment, we also use LLM as the judge and assign a score in the range $[0, 10]$ to each response. Here, we partition the evaluation scores into five intervals: $[0, 2)$, $[2, 4)$, $[4, 6)$, $[6, 8)$, $[8, 10]$, and randomly sample 10 responses from each interval, resulting in 50 responses. Human annotators then rescore these responses on the same 0-10 scale, and we compare the human scores with the LLM scores. Moreover, we also conduct LLM judgment using different LLMs and multiple evaluation runs. Please refer to Appendix D.1 for more details.

Table 6 reports the agreement matrix between LLM and human buckets. For example, the 20% in the first row and second column means that among responses that LLM scores in $[0, 2)$, 20% receive a human score in $[2, 4)$. The results show that for low-quality and high-quality responses, human annotators and the LLM judge agree very well, while for medium-quality responses, the LLM judge tends to be more stringent and humans are slightly more lenient, assigning somewhat higher scores. Overall, the trend is consistent, and both judges rank clearly good and clearly bad responses

Table 6: Bucket-level agreement between human and LLM judgment (rows: LLM buckets, columns: human buckets).

|  | [0,2) | [2,4) | [4,6) | [6,8) | [8,10] |
|---|---|---|---|---|---|
| **[0,2)** | 80% | 20% | 0% | 0% | 0% |
| **[2,4)** | 0% | 20% | 70% | 10% | 0% |
| **[4,6)** | 0% | 0% | 30% | 70% | 0% |
| **[6,8)** | 0% | 0% | 0% | 10% | 90% |
| **[8,10]** | 0% | 0% | 0% | 0% | 100% |

similarly. For all 50 samples, the overall Pearson correlation between LLM scores and human scores is 0.9195, and 92% of responses satisfy | LLM Score - Human Score | $\leq 2$. These statistics indicate strong linear agreement between LLM and human judgments and show that most per-response disagreements are small in magnitude, which supports the use of the LLM judge on this dataset.

## 5 CONCLUSION

In this work, we introduce VeriWeb, a carefully designed, human-annotated benchmark created to address the growing need for verifiable, long-chain web benchmarks in information-seeking agents. Unlike prior benchmarks that focus on single-fact retrieval and outcome-only validation, VeriWeb emphasizes *long-chain complexity* and *subtask-level verifiability*, supporting the development and evaluation of agent capabilities in real-world search workflows. Extensive experiments across a range of leading agent models highlight persistent challenges in large-scale retrieval and synthesis, underscoring the importance of benchmarks like VeriWeb in pushing the frontier of generalist agent intelligence. Future work will focus on scaling VeriWeb with broader data coverage and exploring its role as a training dataset for more robust agent models. We hope VeriWeb serves as a valuable resource for the community, fostering further research into agentic information-seeking.

ETHICS STATEMENT

This paper introduces the VeriWeb benchmark to advance the evaluation and development of information-seeking web agents. While such technologies may have broader societal impacts, including potential misuse for biased retrieval or disinformation, our work is limited to building a human-curated evaluation dataset with a focus on factuality and verifiability. We therefore see no foreseeable ethical concerns or violations of the ICLR Code of Ethics.

REPRODUCIBILITY STATEMENT

To ensure reproducibility, we clearly present the experimental setting in Section 4.1 and Appendix C. Besides, we provide a detailed description of both task instruction construction and human demonstration collection in Appendix A and B. Anonymous code and data are available in the supplementary materials. Since the human demonstration data exceeds 100GB, only one sample is provided for reference due to the file size limit.

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

# Appendix

TABLE OF CONTENTS

# A    TASK INSTRUCTION CONSTRUCTION

To generate realistic and executable instructions, we develop a multi-stage pipeline combining human curation with language model generation, as shown in the left part of Figure 2. Initially, a small batch of seed instructions is manually selected for each topical domain. These seed instructions, representing high-level user intents, are input to a language model to generate a large number of candidate tasks. Human annotators then review these outputs, selecting only those that are grammatically clear, semantically meaningful, and practically feasible. Once a vetted pool of main tasks is established, the language model is prompted to perform subtask decomposition to obtain complete task instructions, including detailed sub-instructions of each subtask. This process is guided by seed instructions and strict formatting constraints. In our early experiments, we primarily used GPT-4.1. After GPT-5 was released, we switched to GPT-5 for generating and decomposing the tasks. Detailed prompts are provided as follows:

---

**Task Instruction Generation Prompt**

You are a professional dataset designer. Your task is to create a dataset for training a web agent. The dataset should require the agent to complete a series of tasks using only the web.

Dataset characteristics:
- The tasks should be high in difficulty, requiring multi-step retrieval, reasoning, and calculation (long-chain tasks).
- Intermediate and final results must be strictly verifiable.
- Answers must be objective and uniquely determined.
- All necessary data must be accessible via publicly available web resources and allow completion within a reasonable time.
- The tasks should have a degree of innovation and diversity.

Format example:
Please identify, among all Nobel Prize laureates in Physics, Chemistry, and Physiology or Medicine from 2019 to 2024, the scholar who published the largest number of papers within the five years prior to their award. Provide the scholar's name, award year, affiliated institution, total number of papers published within those five years, the title of the most cited paper during that period, and the first sentence of that paper's abstract in the original language.

The dataset will be divided into five categories:
1. Science and Academic Research
2. Business, Finance, and Economics
3. Technology and Industrial Innovation
4. Culture, Arts, Entertainment, and Sports
5. Society, Policy, and Sustainable Development

Please design 10 high-quality tasks in the Science and Academic Research category that meet the criteria of being challenging, long-chain, strictly verifiable, feasible, and innovative.

To prevent overly broad tasks, set reasonable constraints—for example, instead of "Find the country with the largest GDP decline from 2016 to 2024," use "Find the G20 country with the largest GDP decline between 2020 and 2024."

Reference examples for Science and Academic Research tasks:
{task1, task2, task3}

---

---

**Task Instruction Decomposition Prompt**

You are a senior dataset designer. Refer to the example format and, based on the five input "overall tasks," automatically break each task down into a list of trajectory subtasks (the decomposition must be reasonable, with consistent style and structure). The result format is a JSON array, where each item is:

```
{{
    "instruct": Overall task description (use the original text of the input overall task),
    "trajectory": [
        {{
            "subtask": Subtask number (consecutive starting from 1),
            "instruct": Specific instruction for the subtask (actionable and verifiable),
            "verification": {{
                "type": "text_match" or "model_verification",
                "check": Precise checkpoint description
            }},
            "steps": []
        }}, ...
    ]
}}
```

Sample data (for structure and style reference only):
{fewshot_str}

Strictly follow the above format and generate complete data (a JSON array) for the following five "overall tasks":
{tasks_json_str}

---

After generation, each batch of instructions undergoes automated filtering, followed by a second, stricter verification phase involving multiple passes of model-based validation. Only those tasks that pass all verification rounds are retained. This procedure enables efficient instruction generation while maintaining the factual correctness, diversity, and task feasibility necessary for web datasets.

## A.1 AUTOMATED FILTERING

The automated filtering stage is a purely rule-based procedure implemented in code and does not invoke any large language model. Its goal is to remove obviously malformed or trivial outputs before any model-based validation. Concretely:

**Format and schema checks.** We parse the model outputs as JSON and enforce a fixed schema. Each entry must be a JSON object containing exactly the fields "instruct" and "subtasks". The "instruct" field must be a non-empty string. The "subtasks" field must be a list whose elements are subtask objects. Each subtask must contain all required fields, each with the correct type, and no additional fields are allowed. Entries that cannot be parsed as valid JSON, that contain missing or extra fields, or that have empty required fields are discarded at this stage.

**Simple difficulty filtering.** In addition to syntax and type checks, we apply simple structural heuristics as a coarse proxy for difficulty. In particular, tasks with too few subtasks (for example, a single-step instruction) are filtered out automatically because they are unlikely to represent the multi-step, web-oriented behavior that we target. This filtering is implemented via deterministic rules on the number of subtasks.

Only instructions that satisfy all of these deterministic checks are passed to the second stage.

## A.2 MODEL-BASED VALIDATION

The second stage is a model-based validation that operates only on the subset of instructions that have passed the automated filtering described above. This is the only stage that uses prompts and large language models. We use a strong LLM (In our early experiments, we primarily used GPT-4.1.

After GPT-5 was released, we switched to GPT-5.) as an automatic reviewer and instruct it via a detailed prompt to evaluate each candidate instruction with respect to several criteria:

---

**Model-based Validation Prompt**

Your role and objective
- You are a high-standard web agent dataset reviewer. Your task is to conduct a strict dual review of structure and content for the input batch and provide a verdict of "Pass", "Revise", or "Reject" for each task.
- Your review scope includes: format and structural compliance, task difficulty and long-chain nature, strict verifiability and uniqueness, feasibility, innovation and diversity, and the reasonableness of scope limitations.
- Do not attempt to execute the tasks or generate answers; only perform review and scoring. Provide clear, actionable revision suggestions for fixable issues.

Input
- The input is a JSON array string {batch_str}.
- Each element must be an object containing the fields:
  - instruct: a non-empty string describing the main task.
  - trajectory: an array containing at least several subtasks; each subtask includes:
    - subtask: an integer starting from 1 and increasing sequentially.
    - instruct: a non-empty string describing the specific instruction for the subtask.
    - verification: a JSON object containing:
      - type: must be only "text_match" or "model_verification".
      - check: a non-empty string that must specify clear, verifiable criteria.
    - steps: must be an empty array [].
- No additional fields are allowed; the JSON must be valid.

Checks to perform
1) Structure and format compliance
- The top level must be a valid JSON array.
- Each entry may only contain the fields instruct and trajectory; no other fields are allowed.
- The trajectory must be an array and include at least 3 subtasks; for long-chain tasks, at least 4 subtasks are recommended, otherwise the difficulty may be insufficient.
- Each subtask must include subtask, instruct, verification, and steps; their types and requirements must conform to the above.
- verification.type must be "text_match" or "model_verification"; verification.check must be a non-empty string.
- If structure or types are non-compliant, mark as "Reject".

2) Task difficulty and long-chain nature
- The main task must be high difficulty, requiring multi-step retrieval and reasoning/computation as a long-chain task; tasks that merely list a single fact or directly look up a single piece of data are not acceptable.
- It should satisfy most of the following conditions:
  - Requires cross-source retrieval (e.g., multiple websites or data sources) and integration, comparison, or computation.
  - Includes time ranges, set constraints (such as G20, a specific database, a designated set of journals), with explicit filtering conditions.
  - Involves non-trivial aggregation, sorting, filtering, or constraints (e.g., filter first, then compare, then perform derived calculations).
  - Requires that intermediate and final results be unique and verifiable, and when necessary, include clear tie-breaking rules for cases of ties.
- If there are fewer than 4 subtasks or the overall task is a simple retrieval with no reasoning required, generally judge as "Revise" (can be improved by adding constraints and steps).

3) Verifiability and uniqueness
- The outputs of the main task and each subtask must be objective, fixed answers or uniquely

---

determinable by clear rules.
- verification.check must provide explicit, executable criteria:
  - For text_match: it should be the expected text or a determinable normalized match (such as an exact string or an explicit regular expression). Do not use vague descriptions.
  - For model_verification: articulate the verification logic (e.g., "compare set equality," "value within $\pm 0.5\%$," "ranking consistently identical"). Do not use generic statements such as "check if correct".
- For tasks that may have ties (most/least/ranking), strict and unambiguous tie-breaking rules must be set in the main task or subtasks. If absent, judge "Revise".
- Specify output units, exchange rates, inflation basis, time zones, date boundaries, etc., to avoid interpretive ambiguity.

4) Feasibility (achievable via public web resources)
- The required data should be reasonably obtainable through publicly accessible web resources (e.g., official websites, authoritative databases). If it requires payment, login, or sources with insufficient credibility, judge "Revise" or "Reject".
- The time range and data scale should be appropriate, avoiding excessive breadth that makes the task inoperable. If the scope is too broad (e.g., "complete statistics for all papers from all universities worldwide with real-time citation data"), suggest narrowing to a reasonable set (e.g., "in OpenAlex, a specified set of journals and time range").
- If the main task contains requirements that are unobtainable or logically self-contradictory (e.g., "verify unpublished data"), judge "Reject".

5) Innovation and diversity
- The task should have some innovation, composite requirements, or complex constraints; it should not be highly repetitive with common templates.
- There should be no obvious duplicate tasks within the batch; if duplicates exist, mark "Reject".

6) Reasonableness of scope limitations
- Tasks should have reasonable scope limitations to prevent the search space from being too large. Excessive breadth warrants "Revise".
- Do not over-restrict to the point that the task loses its challenge (e.g., narrowing to a simple check of a single entity). If the task is evidently reduced to simple retrieval, judge "Revise".

Decision logic:
- Reject: invalid JSON; key fields missing or type errors that are not easily fixable; task inherently infeasible or self-contradictory; requires subjective judgment that cannot be converted to objective standards, etc.
- Revise: can be fixed by adding or refining the scope, adding tie-breaking rules, improving verification.check, adding steps, clarifying data sources, etc.
- Pass: meets all requirements with no major issues.

Output format (strictly follow)
- The output must be a strict JSON array. Each object in the array corresponds to the entry at the same index in the input (index starting from 1), and contains the following fields:
  - index: integer, the sequence number of the entry in the input array (starting from 1).
  - verdict: only "Pass," "Revise," or "Reject".
  - format_valid: Boolean value indicating whether structure and format are fully compliant.
  - issues: an array of strings listing the problems found (empty array if none).
  - reasoning: a brief explanation of the basis for the verdict.
  - revision_suggestions: when the verdict is "Revise," provide specific, actionable modification suggestions; otherwise, an empty string.
- No fields other than those listed above are allowed.
- If the overall input is not a valid JSON array, output a single string: "INVALID_INPUT:

> not a valid JSON array".
>
> Review style
> - Strict, concise, and actionable. Do not output any text unrelated to the above format.
> - Do not execute tasks or generate data; only perform review and provide suggestions.
>
> The data to be checked is as follows:
> {batch_str}

We run 3 model-based validation rounds for each instance. The motivation is to reduce the probability that a low-quality instance passes due to a single stochastic error of the reviewer model. Only instances that are consistently labeled as "Pass" across all rounds survive this stage.

## B  HUMAN DEMONSTRATION COLLECTION

We partnered with a professional data annotation company that follows standardized procedures and internal quality audits. Specifically, our human demonstration collection follows a four-stage adversarial annotation protocol:

**Stage 1: Initial annotation.** We recruited 30 human annotators, all with at least a bachelor's degree, at a cost of $50 per annotation. Human annotators complete the task following a standardized data collection document in Appendix B.2. They conduct keyword-based searches on credible sources, record the entire retrieval process step by step without using any AI tools, and follow requirements on evidence highlighting, screen recording quality, and structured summarization of subtask-level answers. Each annotation session generally lasts 1 to 2 hours. These initial trajectories form the base demonstrations.

**Stage 2: Adversarial dual review.** Each initial trajectory is then examined by 2 independent review teams, with 2 reviewers per team and all reviewers holding at least a bachelor's degree. Reviewers also follow a standardized data review document in Appendix B.3 that covers search logic, evidence consistency, accuracy checks, and compliance with recording constraints. They verify that each retrieval step matches the corresponding sub-instruction, that recorded pages support the submitted answers, and that no prohibited operations occur. The two teams do not see each other's comments, which creates an adversarial, multi-perspective inspection of long, multi-step trajectories that is designed to stress test the process rather than only verify the final answer.

**Stage 3: Iterative refinement.** The original annotator receives consolidated feedback from both review teams and revises the trajectory accordingly. If the trajectory does not fully satisfy the requirements, it is returned for further revision and rechecked. In practice, most items are accepted after no more than 2 refinement rounds. This iterative loop, combined with shared guidelines, helps align annotators on step granularity and reasoning style, which in turn improves cross-sample consistency for these long trajectories.

**Stage 4: Final verification.** A separate reviewer, who did not participate in the earlier stages, performs a final pass before the example is admitted into the released dataset. This reviewer validates correctness, compliance with all document requirements, internal consistency between recorded evidence and submitted answers, and the absence of problematic shortcuts or unfair treatment. Items that fail any requirement are rejected.

Moreover, we employ a game-theoretic incentive mechanism to encourage deep and careful review. The two review teams share a fixed bonus pool ($20 per review) with a high variance asymmetric allocation: 80% of the bonus is awarded to the team whose feedback is judged more comprehensive, actionable, and insightful, and 20% to the other team. This design creates strong incentives for reviewers to maximize review depth and precision instead of free riding, and it rewards the identification of subtle errors, omissions, or potential sources of bias.

Overall, this pipeline (1) encourages exhaustive error detection through adversarial, multi-team scrutiny, (2) mitigates individual annotator bias via independent reviews and a separate final verifier, and (3) enforces consistency through standardized protocols, iterative refinement, and incentives aligned with review quality.

## B.1 ACTION SPACE

For human demonstrations, the action space $\mathcal{A}$ defines a unified set of web operations applicable across web tasks, as shown in Table 7. These actions cover common interaction modalities such as clicks, input, and key events. During execution, the agent selects one action per step from this action set. In some cases, the `result_state()` action is used by the model to output the final result. The specific mapping between the actions recorded during data collection and the web actions is provided as follows.

Table 7: Human Action Space in VeriWeb

| Action Type | Description |
|---|---|
| `left_click([x, y])` | Left-click at given coordinates |
| `right_click([x, y])` | Right-click at given coordinates |
| `drag([x, y])` | Drag at given coordinates |
| `scroll()` | Scroll vertically |
| `input(text)` | Type a string input |
| `key_down(key)` | Press down a single key |
| `result_state()` | Output result statement |

We develop a screen capture tool to support human annotators in collecting detailed task trajectories. Each recorded trajectory logs all mouse and keyboard events, which can be systematically mapped to the predefined action space. The mapping is as follows:

- Scroll wheel events (`WHEEL`) are mapped to the `scroll` action.
- Key press events (`KEY_DOWN`) are mapped to the `key_down` action.
- Text input events (`INPUT`) are mapped to the `input` action.
- Text output events (`RESULT_STATE`) are mapped to the `result_state` action.
- Right-click context menu events (`CONTEXT_MENU`) are mapped to the `right_click` action.
- Tab switching events (`TAB_CHANGE`) are interpreted to the `left_click` action at the corresponding coordinates.
- Mouse drag actions (`MOUSE_DRAG`) are mapped to the `drag` action.
- If a `MOUSE_DOWN` event is not followed by a `MOUSE_DRAG` event, it is interpreted as the `left_click` action.
- Additionally, `MOUSE_UP` events are recorded to help determine the end of drag actions or validate click completions, although they are not directly mapped to any action in the defined space.

This mapping ensures consistency between the raw recorded interactions and the unified action space $\mathcal{A}$, enabling accurate interpretation and reproduction of user behaviors by the model during both training and inference.

## B.2 DATA COLLECTION DOCUMENT

This section outlines the Standard Operating Procedure (SOP) for data collection tasks.

---

**Data Collection Document**

TASK CONTENT
- Based on the instruction tasks provided by the platform, find relevant results and complete a step-by-step recording of all search processes.
- We have provided subtask steps that need to guide the retrieval operations according to these steps.

TASK WORKFLOW
- Claim tasks within the platform.
- Answer sub-instruction questions step by step, ensuring that answers are obtained from reliable data sources found through keyword searches related to the sub-instructions. Ap-

---

propriate modification/merging of sub-instruction content is allowed to adjust search logic (modified sub-instructions must be complete sentences).

- Summarize all sub-instruction answers and fill in the required answer content for the instruction.

- Open the plugin to start recording the task retrieval process. No AI tools or large language models are allowed during recording

- After completing the recording, fill in the corresponding results and sub-results in English.

COLLECTION REQUIREMENTS

**Content Information Requirements**

- Ensure that retrieved answer information corresponds to instruction/sub-instruction content.

- When modifying instructions/sub-instructions, the modified content must be a complete sentence.

- The recorded screen must show relevant data required by instructions/sub-instructions. For data analysis, the integration process must be demonstrated:
  - When relevant data is scattered across different tables, all table data screens need to be retrieved and scrolled through for recording.
  - Extract relevant page information and place it in an online Excel spreadsheet, recording the data processing procedure.

**Recording Detail Requirements**

- No large language models are allowed for finding answers during recording.

- Ensure the recording is entirely in English, including browser language and input content.

- Ensure normal retrieval flow during recording without irrelevant search operations.

- Content that needs to be input during recording can be directly copied and pasted.

- Before recording each sub-instruction, create a browser with only one page at google.com. When searching questions, enter google.com in the browser's URL bar to access the Google interface for searching. Direct searching in the top red URL bar is prohibited.

- When relevant answer information is found during recording, it must be highlighted with the mouse.

- Ensure clean and concise recording screen (shortcut key: Ctrl+Shift+N). Recording screen cannot have other plugin pop-ups (e.g., translation software pop-ups, though small amounts of advertising are allowed).

- Ctrl+F search operations are not allowed during recording.

**Search Source Credibility Hierarchy**
Data retrieval logic priority decreases from top to bottom:

- Directly related enterprise/brand official websites.

- Global organization/government official websites and database websites.

- Wikipedia.

- Niche database official websites.

- News reports.

- Other websites.

## B.3 DATA REVIEW DOCUMENT

This section outlines the Standard Operating Procedure (SOP) for data review tasks.

## Data Review Document

**Review Principles**

- Ensure data answer accuracy.
- Ensure no illogical search processes appear.
- Ensure no Chinese appears (excluding cases where data itself requires searching for Chinese answers, even so, the search process must only use English).

**Review Recording Logic**

- Cannot directly search for answers to reach conclusions; the recording task environment involves searching and recording under unknown conditions.
- Ensure recorded answers correspond to instruction/subtask content; data serves instructions rather than randomly recording irrelevant information.
- Information in corresponding web pages browsed in the video contradicts the given results (e.g., task requires counting occurrences of certain data, but results are inconsistent with comprehensive statistical data on web pages).
- If video cannot confirm whether browsed web page information can support results, verification through links to corresponding web pages is required.

**Review Video Content**

- Unless required by task, recorded web pages cannot contain Chinese (except when instructions require querying Chinese results, but search recording must still use English).
- No operations in the top address bar.
- Cannot directly search for answer results (recording without logic).
- Cannot show large amounts of small advertisements (more than 3 advertisements on the same page is considered large amounts).
- Cannot expose bookmark bars, other irrelevant plugins (including translation plugins), etc.
- Interface may show large language models, but cannot reference/use large language model answers during recording.
- Key information found must be highlighted with mouse selection (if any sub-answer that doesn't require calculation is not highlighted, this item fails).

**Review Text Content**

- Replace corresponding demonstrative pronouns with proper nouns in sub-instructions.
- All groups must be merged; results must be properly grouped.
- Answers must be accurately grouped according to quantity.
- Confirm whether answers requiring counting are correct.
- Confirm whether calculated growth, ratios, differences, and other data are correct.

- Top URL bar only allows entering google.com. All other searches must be conducted within the Google search box.
- AI summary content (Google's built-in Gemini) and directly clicking related web pages are not allowed.
- Based on actual data information obtained and search logic, modify and adjust instruction/subtask field content. For example:
  - When pronouns involve one or two pieces of information, directly replace the pronoun "the country" with the specific country found.
  - When pronouns involve large amounts of information, directly add English parentheses after the pronoun "the country" and fill in corresponding information.
- If instructions require finding "bond name, issuance amount, underwriter, issuing company, use of proceeds, bond rating, and issuance date" for certain content, results can be filled in 7 lines, with each answer separated by "English comma + space".

## C  DETAILED EXPERIMENTAL SETTINGS

### C.1  AGENT PROMPT

The agent prompts for different agents shown below:

---

**Deep Research Agent Prompt**

{question}

---

**Search Engine Agent Prompt**

You are the EXECUTOR agent. You will receive one task description at a time. Your role is to complete the task efficiently, using available tools via function calls when necessary.

Guidelines:
- Always think step by step before responding.
- Provide concise answers.
- If a tool is needed, respond only with the function call — no extra text.
- When the task is complete, respond with: FINAL ANSWER: [your answer here]

---

**Browser-Use Agent Prompt**

We follow the official agent prompt from Browser-Use (Müller & Žunič, 2024).

---

**OWL Agent Prompt**

You are a helpful assistant that can search the web, extract webpage content, simulate browser actions, and provide relevant information to solve the given task.

You are now working in 'working_dir'. All your work related to local operations should be done in that directory.

### Mandatory Instructions
1. **Take Detailed Notes**: You MUST use the 'append_note' tool to record your findings. Ensure notes are detailed, well-organized, and include source URLs. Do not overwrite notes unless summarizing; append new information. Your notes are crucial for the Document Agent.

### Web Search Workflow
1. **Initial Search**: Start with a search engine like 'search_google' or 'search_bing' to get a list of relevant URLs for your research if available.
2. **Browser-Based Exploration**: Use the rich browser toolset to investigate websites.
- **Navigation**: Use 'visit_page' to open a URL. Navigate with 'click', 'back', and 'forward'. Manage multiple pages with 'switch_tab'.
- **Analysis**: Use 'get_som_screenshot' to understand the page layout and identify interactive elements. Since this is a heavy operation, only use it when visual analysis is necessary.
- **Interaction**: Use 'type' to fill out forms and 'enter' to submit.
3. **Detailed Content Extraction**: Prioritize using the scraping tools from 'Crawl4AIToolkit' for in-depth information gathering from awebpage.

### Guidelines and Best Practices
- **URL Integrity**: You MUST only use URLs from trusted sources (e.g., search engine results or links on visited pages). NEVER invent or guess URLs.
- **Thoroughness**: If a search query is complex, break it down. If a snippet is unhelpful but the URL seems authoritative, visit the page. Check subpages for more information.

> - **Persistence**: If one method fails, try another. Combine search, scraper, and browser tools for comprehensive information gathering.
> - **Collaboration**: Communicate with other agents using 'send_message' when you need help. Use 'list_available_agents' to see who is available.
> - **Clarity**: In your response, you should mention the URLs you have visited and processed.

## C.2 LLM-as-a-Judge Prompt

For web tasks, the goal is defined as obtaining a correct textual answer through multi-turn information retrieval and reasoning. Thus, we use GPT-4.1 as a judge to semantically evaluate the correctness of agents' final answers based on the question, ground truth, and model response, and report the LLM-as-a-Judge score. The detailed evaluation prompt is provided as follows:

---

**LLM-as-a-Judge Prompt for Web Task**

You are a strict evaluator assessing answer correctness. You must score the model's prediction on a scale from 0 to 10, where 0 represents an entirely incorrect answer and 10 indicates a highly correct answer.

\# Input
Question:
```
{question}
```

Ground Truth Answer:
```
{answer}
```

Model Prediction:
```
{pred}
```

\# Evaluation Rules
- The model prediction may contain the reasoning process, you should spot the final answer from it.
- Assign a high score if the prediction matches the answer semantically, considering variations in format.
- Deduct points for partially correct answers or those with incorrect additional information.
- Ignore minor differences in formatting, capitalization, or spacing since the model may explain in a different way.
- Treat numerical answers as correct if they match within reasonable precision
- For questions requiring units, both value and unit must be correct

\# Scoring Guide
Provide a single integer from 0 to 10 to reflect your judgment of the answer's correctness.

\# Strict Output format example
4

---

We conduct a quantitative breakdown of different failure modes using GPT-5 as a judge, including (a) misinformation, (b) incomplete result, (c) retrieval failure, and (d) irrelevant result. The detailed evaluation prompt is provided as follows:

**LLM-as-a-Judge Prompt for Failure Modes**

You are an expert judge analyzing failure modes of answers produced by research agents.

Your task: given (1) the original user instruction, (2) the ground-truth reference answer, and (3) the agent's prediction, identify which of the following failure modes the prediction exhibits. You may select multiple failure modes or none.

Failure mode definitions (names only):
- (a) misinformation
- (b) incomplete_result
- (c) retrieval_failure
- (d) irrelevant_result

The categories are not mutually exclusive. If multiple failure modes apply, select all of them. If the prediction is fully correct and complete, you should return an empty list of error_categories.

Output format:
Return a single JSON code block with exactly two fields:
- "reason": a concise English explanation (1–3 sentences) of why you chose the labels.
- "error_categories": an array of zero or more labels selected from:
    ["misinformation", "incomplete_result", "retrieval_failure", "irrelevant_result"].

Few-shot examples (only expected JSON outputs):

Example 1 — single label
```json
{{
"reason": "The prediction is on-topic and broadly correct but only lists two planets instead of the requested three.",
"error_categories": ["incomplete_result"]
}}
```

Example 2 — multiple labels
```json
{{
"reason": "The prediction misses Italy's capital entirely and also misidentifies Germany's capital as Munich.",
"error_categories": ["misinformation", "incomplete_result"]
}}
```

Now classify the following real example.

Instruction:
{instruction}

Reference answer:
{reference_answer}

Model prediction:
{prediction}

Return only the JSON code block.

Table 8: Comparison of the task completion rate of deep research agents on VeriWeb across different LLM judges and multiple evaluation runs.

| Model | Judge | Run 1 | Run 2 | Run 3 | Run 4 | Run 5 | Run 6 | Run 7 | Run 8 | Average |
|---|---|---|---|---|---|---|---|---|---|---|
| Gemini-2.5-Flash | OpenAI-o3 | 31.16 | 30.17 | 30.79 | 30.93 | 30.23 | 31.03 | 30.63 | 30.79 | $30.72 \pm 0.33$ |
| | GPT-5 | 29.40 | 29.14 | 29.07 | 29.64 | 29.17 | 29.54 | 29.67 | 29.54 | $29.40 \pm 0.22$ |
| Gemini-2.5-Pro | OpenAI-o3 | 33.94 | 33.97 | 34.34 | 34.30 | 34.07 | 34.14 | 35.26 | 33.54 | $34.20 \pm 0.47$ |
| | GPT-5 | 32.78 | 32.95 | 33.11 | 33.64 | 33.25 | 33.31 | 33.11 | 33.28 | $33.18 \pm 0.24$ |
| OpenAI-o4-mini | OpenAI-o3 | 25.93 | 25.89 | 26.03 | 25.86 | 25.86 | 25.99 | 26.03 | 26.09 | $25.96 \pm 0.08$ |
| | GPT-5 | 21.23 | 20.03 | 20.43 | 20.96 | 20.60 | 22.05 | 21.72 | 20.99 | $21.00 \pm 0.62$ |
| OpenAI-o3 | OpenAI-o3 | 36.26 | 36.03 | 37.19 | 36.72 | 36.13 | 35.86 | 35.79 | 36.23 | $36.28 \pm 0.44$ |
| | GPT-5 | 36.03 | 35.07 | 35.93 | 36.62 | 35.96 | 35.66 | 36.23 | 36.16 | $35.96 \pm 0.42$ |

# D  DETAILED TASK ANALYSIS

## D.1  LLM-AS-A-JUDGE AGREEMENT

In the original experiments, we use OpenAI-o3 as the judge. Here, we have additionally used GPT-5 as another judge and repeated each evaluation eight times per judge to measure both cross-judge agreement and stability under resampling. Due to time constraints, we conduct this analysis on the deep research agent with different backbone models. Table 8 reports the task completion rate for each model under each judge and run, together with the average and standard deviation over 8 runs. We observe that (1) For most models, the average scores under OpenAI-o3 and GPT-5 differ by at most about 2%, showing high consistency across judges. The only exception is OpenAI-o4-mini, where GPT-5 assigns a lower average score by about 5%. Manual inspection suggests that GPT-5 is particularly conservative on OpenAI-o4-mini outputs that contain the correct answer mixed with irrelevant or noisy content, and often treats such responses as incorrect. Importantly, regardless of which judge we use, the ranking of models remains stable and always satisfies OpenAI-o3 > Gemini-2.5-Pro > Gemini-2.5-Flash > OpenAI-o4-mini. (2) For each judge, the standard deviation across 8 runs is within about 1%, which shows that repeated sampling of the LLM judge yields very similar aggregate scores and that the evaluation is stable.

## D.2  HUMAN PERFORMANCE

In the VeriWeb benchmark, we regard human performance as essentially at ceiling (very close to perfect), because all ground truth answers in our dataset are indeed given by human annotators. As discussed in our paper, our benchmark targets large-scale retrieval and synthesis of information from diverse sources. These are exactly the kinds of tasks where current agents still underperform, yet humans routinely solve them in everyday and professional settings. Therefore, to construct a reliable reference, all task answers are produced by human experts rather than models.

Moreover, as stated in Appendix B, we recruited human annotators to provide the answers that serve as the gold standard ground truth. Each annotation session typically lasts 1 to 2 hours. After the initial annotation, the internal review teams will check whether each sample satisfies the task requirements. Samples that do not fully meet the requirements are sent back for revision and then re-checked, and in practice most items are approved after no more than two rounds of revision. Given this process, we expect human performance on these tasks to be very close to 100%.

To more directly address the concern about a human performance baseline, we have additionally designed an experiment to measure human performance under a time limit (12 minutes per task, which roughly matches the typical completion time of the deep research agents and browser-use agents). Based on the task difficulty levels in our original experiment, we randomly sampled 10 tasks from each difficulty level and recruited 5 additional human annotators (all with at least a bachelor's degree, at a cost of $20 per annotation) to complete them under this time limit. In this setting, humans did not successfully complete any tasks, so we report only task completion rates. The results in Table 9 show that human performance drops substantially under this constraint. This illustrates a regime where agents can surpass time-limited human performance, even though the ground truth remains defined by unconstrained human experts.

Table 9: Comparison of task completion rates on VeriWeb. Human performance is measured under a 12-minute time limit per task. Deep research agents typically require about 12 minutes per task, search engine agents about 4 minutes per task, browser-use agents about 10 minutes per task, and multi-agent systems about 40 minutes per task.

| Model | Level 1 | Level 2 | Level 3 | Level 4 | Level 5 |
|---|---|---|---|---|---|
| *Deep Research Agents* | | | | | |
| Gemini-2.5-Flash | 93.0 | 47.0 | 24.0 | 1.0 | 0.0 |
| Gemini-2.5-Pro | 84.0 | 56.0 | 8.0 | 6.0 | 0.0 |
| OpenAI-o4-mini | 63.0 | 34.0 | 13.0 | 3.0 | 0.0 |
| OpenAI-o3 | 75.0 | 63.0 | 8.0 | 7.0 | 0.0 |
| *Search Engine Agents* | | | | | |
| OpenAI-o3 | 46.0 | 42.0 | 21.0 | 5.0 | 0.0 |
| GPT-5 | 80.0 | 58.0 | 0.0 | 6.0 | 0.0 |
| *Browser-Use Agents* | | | | | |
| Qwen-VL-Max | 17.0 | 12.0 | 0.0 | 0.0 | 0.0 |
| DeepSeek-V3.1 | 23.0 | 38.0 | 4.0 | 4.0 | 0.0 |
| Gemini-2.5-Flash | 40.0 | 41.0 | 7.0 | 1.0 | 1.0 |
| Gemini-2.5-Pro | 45.0 | 35.0 | 16.0 | 4.0 | 4.0 |
| Claude-3.7-Sonnet | 78.0 | 31.0 | 16.0 | 5.0 | 3.0 |
| Claude-4.0-Sonnet | 42.0 | 16.0 | 4.0 | 2.0 | 0.0 |
| OpenAI-o3 | 62.0 | 47.0 | 5.0 | 5.0 | 2.0 |
| GPT-4o | 18.0 | 15.0 | 2.0 | 0.0 | 0.0 |
| GPT-4.1 | 57.0 | 35.0 | 10.0 | 3.0 | 4.0 |
| GPT-5 | 34.0 | 35.0 | 5.0 | 3.0 | 0.0 |
| *Multi-Agent Systems* | | | | | |
| Qwen3-235B | 17.0 | 14.0 | 1.0 | 0.0 | 0.0 |
| DeepSeek-V3.1 | 43.0 | 6.0 | 0.0 | 0.0 | 0.0 |
| OpenAI-o3 | 69.0 | 62.0 | 6.0 | 4.0 | 0.0 |
| GPT-5 | 17.0 | 15.0 | 9.0 | 0.0 | 0.0 |
| **Human** | 47.0 | 40.0 | 15.0 | 6.0 | 1.0 |

## D.3 TASK EXAMPLE

The web tasks focus on deep research requiring multi-turn information retrieval and reasoning. In VeriWeb, these tasks span five key thematic domains: scientific and academic research; finance and economics; technology and innovation; arts and entertainment; and social policy and sustainability. Below are some examples of web tasks.

---

**Web Task Example - Scientific and Academic Research**

**Task Instruction**
Identify the earliest known warship that sank on its maiden voyage. Provide the vessel's commonly accepted name, estimated sinking year or century, salvage year, location described by sea area and the exclusive economic zone of the country it lies in, as well as the museum currently displaying the wreck and the official name of any anchor-related artifacts from it in the museum's collection.

**Task Answer**
Shipwreck: Vasa
Year of sinking: 1628
Salvage year: 1961
Location: Stockholm, Sweden
Museum: Vasa Museum
Artifact: Ankarstock

---

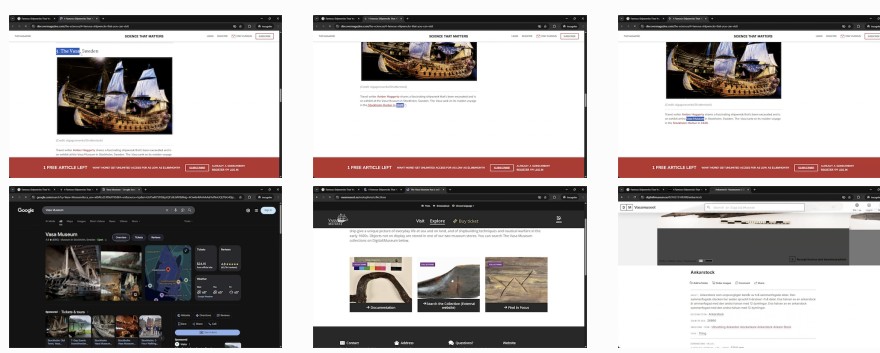

Figure 7: Human demonstration screenshots for the scientific and academic research task example.

**Subtask 1 Instruction**

Compile a list of major shipwreck discoveries where the primary search and identification technology used was multibeam sonar.

**Subtask 1 Answer**

| Name | Year |
|---|---|
| USS Kittiwake | 2011 |
| C-50 Naufragio Vicente Palacio Riva Ship | 2000 |
| The Vasa | 1961 |
| The Lusitania | 1915 |

**Subtask 2 Instruction**

For each shipwreck on the list, find its estimated sinking date (year or century). Identify the wreck with the earliest sinking date.

**Subtask 2 Answer**

| Name | Year |
|---|---:|
| USS Kittiwake | 1994 |
| C-50 Naufragio Vicente Palacio Riva Ship | 2000 |
| The Vasa | 1628 (oldest) |
| The Lusitania | 1906 |

**Subtask 3 Instruction**

Confirm the full name of the organization, or institution responsible for the discovery of the Vasa.

**Subtask 3 Answer**

Organization: Vasa Museum

**Subtask 4 Instruction**

Determine The Vasa's location, specifying the sea or ocean body and the Exclusive Economic Zone (EEZ) of the relevant coastal nation.

**Subtask 4 Answer**

Location: Stockholm, Sweden

**Subtask 5 Instruction**

Search museum databases and archaeological reports to find a museum that currently exhibits The Vasa.

**Subtask 5 Answer**
Museum: Vasa Museum
________________________________________________

**Subtask 6 Instruction**
From the Vasa Museum's official collection catalog or website, find the official name of the specific artifact related to the anchor stock in the museum's collection.

**Subtask 6 Answer**
Name: Ankarstock

## Web Task Example - Finance and Economics

**Task Instruction**
Among all Chinese banks listed in Hong Kong from 2022 to 2023, list the bank with the highest increase in net interest margin ranking, and provide: (1) bank name, (2) net interest margin values before and after the increase, (3) stock code, (4) total asset growth rate, and (5) chairman's name.

**Task Answer**
Bank Name: Hang Seng Bank
Net Interest Margin Values Before and After the Increase: 1.75%, 2.30%
Stock Code: 0011.HK
Total Asset Growth Rate: -8.75%
Chairman's Name: Irene Lee

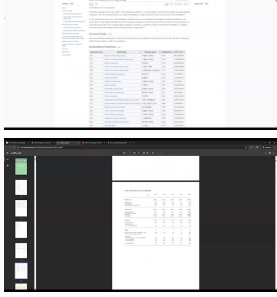 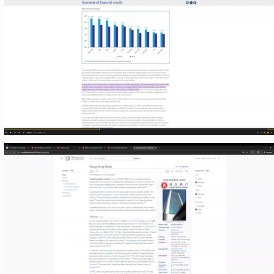 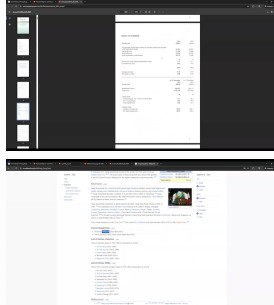

Figure 8: Human demonstration screenshots for the finance and economics task example.
________________________________________________

**Subtask 1 Instruction**
Collect the list of the top 10 licensed banks in Hong Kong by total assets in 2024 and their respective annual net interest margin (NIM) data.

**Subtask 1 Answer**
Hongkong and Shanghai Banking Corporation Limited (The): 10,500,393 HK$ million
Bank of China (Hong Kong) Limited: 3,685,578 HK$ million
Standard Chartered Bank (Hong Kong) Limited: 2,534,695 HK$ million
Hang Seng Bank, Limited: 1,692,094 HK$ million
Industrial and Commercial Bank of China (Asia) Limited: 915,960 HK$ million
Bank of East Asia, Limited (The): 860,361 HK$ million
Nanyang Commercial Bank, Limited: 555,149 HK$ million
China Construction Bank (Asia) Corporation Limited: 493,858 HK$ million
China CITIC Bank International Limited: 470,387 HK$ million
DBS Bank (Hong Kong) Limited: 467,621 HK$ million
________________________________________________

**Subtask 2 Instruction**
Collect the list of the top 10 licensed banks in Hong Kong from 2022 to 2023 and their

annual net interest margin data, and calculate the increase in net interest margin for each bank and identify the bank with the largest increase.

**Subtask 2 Answer**
Bank: Hang Seng Bank
NIM Increase: 55bp

---

**Subtask 3 Instruction**
Find the following for the bank: (1) name, (2) specific net interest margin values for 2022 and 2023, (3) stock code.

**Subtask 3 Answer**
Name: Hang Seng Bank
2022 NIM: 1.75%
2023 NIM: 2.30%
Stock Code: 0011.HK

---

**Subtask 4 Instruction**
Find the bank's(Hang Seng Bank) total asset data for 2022-2024 and calculate the total asset growth rate.

**Subtask 4 Answer**
2022 asset data: 1,854.4 HK$bn
2023 asset data: 1,692.1 HK$bn
Growth rate: -8.75%

---

**Subtask 5 Instruction**
Find the current chairman's name of the bank(Hang Seng Bank).

**Subtask 5 Answer**
Chairman: Irene Lee

---

**Web Task Example - Technology and Innovation**

**Task Instruction**
Identify the pharmaceutical company that had the FDA-approved new molecular entities (NMEs) between 2020 and 2024, where at least one of these drugs achieved blockbuster status (over $1 billion in annual sales) within 24 months of approval. List the company name, total number of NMEs approved, the name and indication of the fastest blockbuster drug, its peak annual sales figure, and the name and specialization of the lead scientist credited with its discovery.

**Task Answer**
Company Name: Pfizer Inc.
Total NME Approvals: 9
Details of the company with the largest number of approvals: Approval date drug trade name drug generic name 2021-11-05 Paxlovid™ nirmatrelvir/ritonavir, 2022-05-25 Cibinqo™ abrocitinib, 2023-01-30 Zavzpret® zavegepant, 2023-05-25 Paxlovid nirmatrelvir/ritonavir, 2023-06-05 Litfulo ritlegepitinib, 2023-08-22 Penbraya™ pentavalent meningococcal, 2023-10-12 Velsipity™ etrasimod, 2024-03-14 Rezdiffra* resmetirom, 2023-03-09 Zavzepant* zavegepant
Fastest Blockbuster Drug: Paxlovid (nirmatrelvir/ritonavir)
Indication: treatment of mild-to-moderate COVID-19 in adults and pediatric patients (12 years of age and older weighing at least 40 kg) who are at high risk for progression to severe COVID-19
Peak Annual Sales: $18.933 billion (2022)
Lead Scientist: Dafydd Owen

Specialization: medicinal chemist in the design and synthesis of drug-like molecules

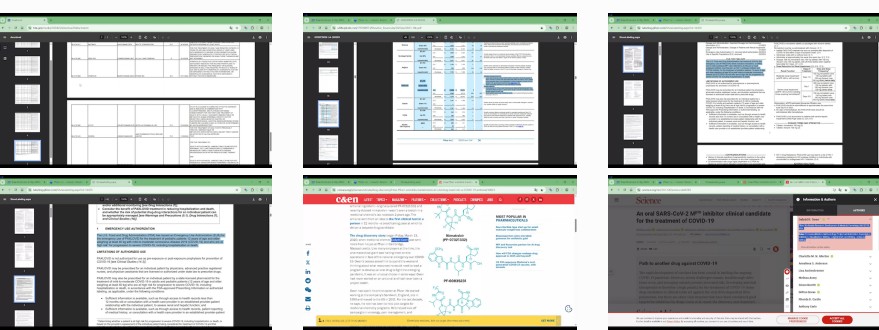

Figure 9: Human demonstration screenshots for the technology and innovation task example.

**Subtask 1 Instruction**

Compile a statistical summary of all FDA-approved NMEs (2020-2024), identify the company with the highest number of approvals, and report its approved drugs with both brand and generic names.

**Subtask 1 Answer**

The ranking of the number of NME approvals of the company:

|   | Company name | Approved quantity |
|---|---|---|
| 1 | Pfizer Inc. | 9 |
| 2 | Novartis Pharmaceuticals | 7 |
| 3 | Bristol Myers Squibb | 6 |
| 4 | Merck Sharp & Dohme | 5 |
| 5 | Takeda Pharmaceuticals | 4 |
| 6 | Eli Lilly and Company | 3 |

Company Name: Pfizer Inc.
Total NME Approvals: 9
Details of the company with the largest number of approvals:

| Approval date | drug Trade Name | Generic Name |
|---|---|---|
| 2021-11-05 | Paxlovid™ | nirmatrelvir/ritonavir |
| 2022-05-25 | Cibinqo™ | abrocitinib |
| 2023-01-30 | Zavzpret® | zavegepant |
| 2023-05-25 | Paxlovid | nirmatrelvir/ritonavir |
| 2023-06-05 | Litfulo | ritlegepitinib |
| 2023-08-22 | Penbraya™ | pentavalent meningococcal |
| 2023-10-12 | Velsipity™ | etrasimod |
| 2024-03-14 | Rezdiffra* | resmetirom |
| 2023-03-09 | Zavzepant* | zavegepant |

**Subtask 2 Instruction**

Among qualifying companies, identify Pfizer Inc. with the most FDA-approved NMEs and find which of their drugs reached blockbuster status fastest after approval and its peak annual sales figure.

**Subtask 2 Answer**

Fastest Blockbuster Drug: Paxlovid (nirmatrelvir/ritonavir)
Peak Annual Sales: $18.933 billion (2022)

**Subtask 3 Instruction**
Find the primary indication for Paxlovid (nirmatrelvir/ritonavir).

**Subtask 3 Answer**
Indication: treatment of mild-to-moderate COVID-19 in adults and pediatric patients (12 years of age and older weighing at least 40 kg) who are at high risk for progression to severe COVID-19

―――――――――――――――――――――――――――――――――――――――――――――――――――

**Subtask 4 Instruction**
Search for the lead scientist or principal investigator credited with discovering Paxlovid (nirmatrelvir/ritonavir), including their full name and area of specialization.

**Subtask 4 Answer**
Lead Scientist: Dafydd Owen
Specialization: medicinal chemist in the design and synthesis of drug-like molecules

---

## Web Task Example - Arts and Entertainment

**Task Instruction**
Identify the film with the highest production cost return ratio (box office/production cost) among all movies that grossed over $1 billion worldwide between 2020 and 2024, and list its title, director, production cost, global box office, main filming location, as well as the name of the highest-level film award it received and the city where the award ceremony was held.

**Task Answer**
Title: The Super Mario Bros. Movie
Director: Aaron Horvath
Production cost: $100,000,000
Global box office: $1,360,847,665
Main filming location: Paris, France
The name of the highest-level film award it received and the city where the award ceremony was held: Festival Film Bandung - Film Impor Terpuji / Commendable Imported Film, Bandung, Indonesia

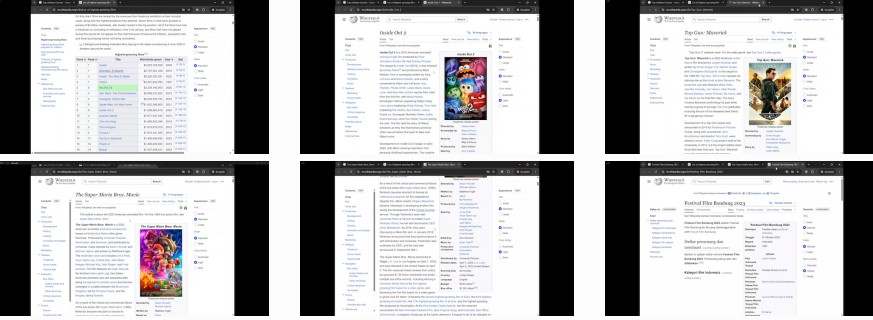

Figure 10: Human demonstration screenshots for the arts and entertainment task example.

―――――――――――――――――――――――――――――――――――――――――――――――――――

**Subtask 1 Instruction**
Collect a list of all films worldwide with box office earnings exceeding $1 billion from 2020 to 2024, along with their box office data.

**Subtask 1 Answer**
Avatar: The Way of Water: $2,320,250,281
Inside Out 2: $1,698,863,816
Spider-Man: No Way Home: $1,922,598,800

Top Gun: Maverick: $1,495,696,292
Barbie: $1,447,038,421
The Super Mario Bros. Movie: $1,360,847,665
Deadpool & Wolverine: $1,338,073,645
Moana 2: $1,059,242,164

---

**Subtask 2 Instruction**
Search the production cost of each film, and calculate the ratio of box office to production cost to identify the film with the highest return on investment. List only the highest-rated movies and their ratios.

**Subtask 2 Answer**
The Super Mario Bros. Movie, 13.61

---

**Subtask 3 Instruction**
Find the director's name of The Super Mario Bros. Movie, the specific production cost, and the exact global box office revenue.

**Subtask 3 Answer**
Aaron Horvath, $100,000,000, $1,360,847,665

---

**Subtask 4 Instruction**
Search for the main filming locations of The Super Mario Bros. Movie.

**Subtask 4 Answer**
Paris, France

---

**Subtask 5 Instruction**
Find all the film awards that The Super Mario Bros. Movie has received, identify the highest-level award among them, and find the host city of the corresponding award ceremony.

**Subtask 5 Answer**
Festival Film Bandung - Film Impor Terpuji / Commendable Imported Film, Bandung, Indonesia

---

**Web Task Example - Social Policy and Sustainability**

**Task Instruction**
Identify the G20 country that achieved the largest percentage decrease in CO2 emissions per capita between 2015 and 2023 while simultaneously recording a real GDP growth of over 20% in the same period. List the country's name, its official head of government as of year-end 2023, the primary renewable energy source by installed capacity, and the official title of its most recent Nationally Determined Contribution (NDC) report submitted to the UNFCCC.

**Task Answer**
The country's name: UK
UK's official head of government as of year-end 2023: Rishi Sunak
The primary renewable energy source by installed capacity: wind sources
The official title of its most recent Nationally Determined Contribution (NDC) report submitted to the UNFCCC: United Kingdom of Great Britain and Northern Ireland's 2035 Nationally Determined Contribution

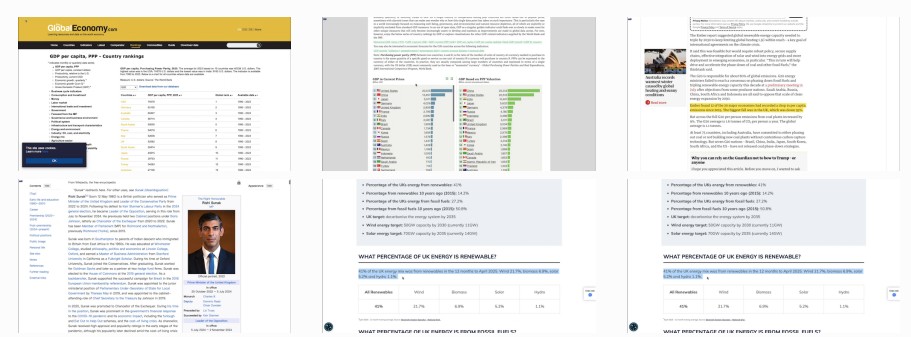

Figure 11: Human demonstration screenshots for the social policy and sustainability task example.

---

**Subtask 1 Instruction**
Identify the G20 country that achieved a real GDP growth of over 20% between 2015 and 2023.

**Subtask 1 Answer**
USA, China, india, UK, Brazil, Russia, Canada, Mexico, Indonesia, Turkey, Saudi Arabia, Argentina

---

**Subtask 2 Instruction**
Identify the country with the largest percentage decrease in CO2 emissions per capita.

**Subtask 2 Answer**
The country name: UK

---

**Subtask 3 Instruction**
Find the full name of the head of government for UK, who was in office on December 31, 2023.

**Subtask 3 Answer**
The full name: Rishi Sunak

---

**Subtask 4 Instruction**
Research the energy profile of UK to determine its primary renewable energy source based on the latest available data for installed capacity (in MW or GW).

**Subtask 4 Answer**
UK's primary renewable energy source: wind sources

---

**Subtask 5 Instruction**
Search the official UNFCCC registry or UK's national environmental ministry website to find its most recently submitted Nationally Determined Contribution (NDC) report. Record its full official title and the year it was published/submitted.

**Subtask 5 Answer**
The official title of its most recent Nationally Determined Contribution (NDC) report submitted to the UNFCCC: United Kingdom of Great Britain and Northern Ireland's 2035 Nationally Determined Contribution

---

## E  ADDITIONAL DISCUSSIONS WITH RELATED WORKS

HotpotQA (Yang et al., 2018) evaluates multi-hop question answering on a static Wikipedia corpus and provides sentence-level supporting fact annotations that indicate the evidence used to answer each final question. However, the standard evaluation focuses on the correctness of the final answer and its associated supporting facts for a single global question, without explicitly defining intermediate subgoals that can be evaluated as separate steps in a longer information-seeking workflow. In contrast, VeriWeb explicitly decomposes each realistic information-seeking task into a sequence of interdependent subtasks, each with its own sub-instruction and answer. Each subtask is designed to serve as a valid entry point, enabling agent evaluation at different stages of the overall task.

TheAgentCompany (Xu et al., 2024), similar to our VeriWeb framework, provides checkpoint-based evaluation at the level of subtask goals. We would like to clarify that our setting differs from TheAgentCompany mainly in the following aspects:

- **Task type.** TheAgentCompany builds a simulated software development company environment with internal websites and data, where agents perform tasks such as software engineering, project management, and financial analysis. In contrast, VeriWeb targets open-ended information seeking in realistic web environments, focusing on long-horizon web tasks that require large-scale retrieval and synthesis of information from diverse real-world websites.

- **Subtask evaluation.** In TheAgentCompany, checkpoints are tightly coupled to the underlying environment state. As a result, it is nontrivial to start the evaluation from an intermediate subtask, because one must restore the entire environment to the corresponding state. In VeriWeb, each subtask can easily serve as an independent starting point, because the ground truth answers to all earlier subtasks are stored as text and can be given to the agent as contextual input.

- **Released artifact.** TheAgentCompany primarily provides a benchmark environment for evaluating LLM agents that act as digital workers, but without releasing related human trajectories. VeriWeb releases a high-cost, human-annotated dataset of 302 verifiable long-chain task trajectories across diverse real-world domains, capturing both long-chain complexity and subtask-level verifiability. On top of this dataset, we designed the VeriWeb benchmark, which supports fine-grained analysis of existing agent capabilities.

## F  THE USE OF LLMS

In this work, LLMs were used exclusively for two auxiliary purposes: (1) LLMs assisted with minor language polishing and stylistic refinement. (2) LLMs were used as auxiliary tools in task construction and preliminary review, while humans designed the framework, curated the data, and performed the final validation. Note that LLMs did not contribute to research ideation, methodology design, experimental design, or data analysis. The authors take full responsibility for all content.

