# OpenReview forum: "VeriWeb: Verifiable Long-Chain Web Benchmark for Agentic Information-Seeking"
_ICLR.cc/2026/Conference — Submitted to ICLR 2026_

### Official Review · Reviewer_tn1J · 2025-10-31

**Soundness:** 2
**Presentation:** 3
**Contribution:** 3
**Rating:** 6
**Confidence:** 3

**Summary:**

This paper introduces a new benchmark for web agents, primarily focusing on information retrieval / deep research tasks. The authors claim that their benchmark is novel and introduces features not previously seen in other benchmarks.
1. Long, multi-question information retrieval - Each sample spans multiple questions that test both breadth and depth of reasoning, requiring agents to perform multi-hop information retrieval while maintaining contextual coherence across web pages.
2. Intermediate step verifiability - A large task broken into several, intermediate subtasks, each of which has verifiable, fixed ground truth answers.

The authors provide detailed analysis of benchmark’s data statistics and discuss the evaluation metrics used to prove the uniqueness and complexity of their benchmark. The authors use these metrics to evaluate several agentic frameworks and frontier models on their benchmark, showing generally low scores across metrics like Success Rate (SR) and Completion Rate (CR). They also provide a comparative analysis of how models and agent paradigms perform across different domains and across breadth vs depth-oriented tasks.

**Strengths:**

1. The proposed benchmark introduces long-horizon, multi-question challenge to deep research agents with verifiable subtasks, where the multi-question consistency feature is an important extension to existing benchmarks that largely focus on single-fact tasks or subjective report generation.

2. Empirical result shows that the benchmark is difficult for  most advanced agentic systems and frontiner models, which provides a a potential new direction the research community can make measurable progress towards.

3. The experiments and analysis are relative extensive, and covers a good representation of agent paradigms and frontier models to make sure that the observations of benchmark difficulty is not overfitting particular agent designs or models.

**Weaknesses:**

1. The main evaluation metrics this paper use are based on LLM as a judge, but there is no study of how well the LLM judge matches human assessment of answer quality on this dataset. This is especially important since this dataset claims to expose reliable subtask-level assessment of agent quality. If the evaluation itself is not stable enough, it puts into question all of the evaluation results presented in this work. One potential alternative, for instance, taken by the GAIA benchmark, is to specify the output format with task instructions and use stricter evaluator functions like string match.

2. One of the main contributions of this benchmark concerns subtask-level verification, which is important and useful for complex information retrieval tasks that is oft-neglected in many benchmarks. However, the paper fails to discuss this contribution in relation to previous work in the literature. For one, HotpotQA (which is cited by this paper) introduced supporting fact evaluation for reasoning, which is a proxy to checking the necessary reasoning steps are reached to arrive at the final answer. As a more recent example, the Agent Company [1] uses subtask rewards to evaluate agents on long-horizon agent tasks.

3. Some claims in the paper are unsubstantiated. For instance, L360 states "Search engine agents, constrained to passive retrieval, typically achieve the lowest success rates." But the agents that achieved the lowest success rates in Table 2 are Browser-Use Agents and Multi-agent Systems.

[1] TheAgentCompany: Benchmarking LLM Agents on Consequential Real World Tasks. (https://arxiv.org/pdf/2412.14161)

**Questions:**

1. Why does the paper use Browser actions as the main metric for step / action efficiency? Just because this is how humans do it with a browser doesn't mean it's the most effective (as evidenced by the results) or efficient way for agents to do it.

2. Some examples in Figure 4 have SR that's 0 < SR < 1, does this mean that SR is not a binary 0/1 metric, or is this averaged over several runs? What does it mean if a task is 50% successful?

---

> ### Author Response · Authors · 2025-11-23
> **Rebuttal (1/3)**
>
> We sincerely appreciate the reviewer for highlighting the strengths of VeriWeb, noting that it `serves as an important extension` to existing benchmarks, `offers a potential new direction` for the research community, and includes relatively `extensive experiments and analysis`. We have carefully revised the manuscript according to your valuable suggestions. Below we address the main points raised in the review.
>
> ---
>
> **[W1] there is no study of how well the LLM judge matches human assessment of answer quality on this dataset.**
>
> Thanks for your insightful comment. Because our task answers contain many items, it is hard to validate them directly via string matching. As suggested, we have additionally conducted experiments comparing human and LLM judgment (using different LLMs and multiple evaluation runs) in order to validate the stability and reliability of our evaluation protocol.
>
> **(1) Human judgment.** Since the deep research agent using OpenAI-o3 performs well in the original experiment, we use its responses for this analysis. In the original experiment, we use LLM as the judge and assign a score in the range $[0, 10]$ to each response. Here, we partition these scores into five intervals, and randomly sample 10 responses from each interval, resulting in 50 responses. Human annotators then rescore these responses, and we compare the human scores with the LLM scores.
>
> Table R1 reports the agreement matrix between LLM and human buckets. For example, the 20\% in the first row and second column means that among responses that LLM scores in $[0,2)$, 20\% receive a human score in $[2,4)$. The results show that **for low-quality and high-quality responses, human annotators and the LLM judge agree very well**, while **for medium-quality responses, the LLM judge tends to be more stringent and humans are slightly more lenient, assigning somewhat higher scores**. Overall, the trend is consistent, and both judges rank clearly good and clearly bad responses similarly. For all 50 samples, the overall Pearson correlation between LLM scores and human scores is 0.9195, and 92% of responses satisfy | LLM Score - Human Score | $\leq$ 2. These statistics indicate strong linear agreement between LLM and human judgments and show that most per-response disagreements are small in magnitude, which supports the use of the LLM judge on this dataset. *These new results have been updated in Section 4.5 (Page 10) of the revised manuscript.*
>
> Table R1 (Table 6 in the revision): Bucket-level agreement between human and LLM judgment (rows: LLM buckets, columns: human buckets).
> ||[0,2)|[2,4)|[4,6)|[6,8)|[8,10]|
> |---|---|---|---|---|---|
> |**[0,2)**|80\%|20\%|0\%|0\%|0\%|
> |**[2,4)**|0\%|20\%|70\%|10\%|0\%|
> |**[4,6)**|0\%|0\%|30\%|70\%|0\%|
> |**[6,8)**|0\%|0\%|0\%|10\%|90\%|
> |**[8,10]**|0\%|0\%|0\%|0\%|100\%|
>
> **(2) LLM judgment using different LLMs and multiple evaluation runs.** In the original experiments, we use OpenAI-o3 as the judge. Here, we use GPT-5 as another judge and repeat each evaluation eight times per judge. Due to time constraints, we conduct this analysis on the deep research agent with different backbone models.
>
> The results in Table R2 show that (1) For most models, the average scores under OpenAI-o3 and GPT-5 differ by at most about 2\%, showing **high consistency across judges**. The only exception is OpenAI-o4-mini, where GPT-5 assigns a lower average score by about 5\%. Manual inspection suggests that GPT-5 is particularly conservative on OpenAI-o4-mini outputs that contain the correct answer mixed with irrelevant or noisy content, and often treats such responses as incorrect. Importantly, regardless of which judge we use, the ranking of models remains stable and always satisfies OpenAI-o3 > Gemini-2.5-Pro > Gemini-2.5-Flash > OpenAI-o4-mini. (2) For each judge, the standard deviation across 8 runs is within about 1\%, which shows that **repeated sampling of the LLM judge yields very similar aggregate scores and that the evaluation is stable**. *These new results have been updated in Appendix D.1 (Page 27) of the revised manuscript.*
>
> Table R2 (Table 8 in the revision): Comparison of the task completion rate of deep research agents across different LLM judges and multiple evaluation runs.
> |Model|Judge|Run 1|Run 2|Run 3|Run 4|Run 5|Run 6|Run 7|Run 8|Average|
> |-|-|-|-|-|-|-|-|-|-|-|
> |Gemini-2.5-Flash|OpenAI-o3|31.16|30.17|30.79|30.93|30.23|31.03|30.63|30.79|30.72$\pm$0.33|
> ||GPT-5|29.40|29.14|29.07|29.64|29.17|29.54|29.67|29.54|29.40$\pm$0.22|
> |Gemini-2.5-Pro|OpenAI-o3|33.94|33.97|34.34|34.30|34.07|34.14|35.26|33.54|34.20$\pm$0.47|
> ||GPT-5|32.78|32.95|33.11|33.64|33.25|33.31|33.11|33.28|33.18$\pm$0.24|
> |OpenAI-o4-mini|OpenAI-o3|25.93|25.89|26.03|25.86|25.86|25.99|26.03|26.09|25.96$\pm$0.08|
> ||GPT-5|21.23|20.03|20.43|20.96|20.60|22.05|21.72|20.99|21.00$\pm$0.62|
> |OpenAI-o3|OpenAI-o3|36.26|36.03|37.19|36.72|36.13|35.86|35.79|36.23|36.28$\pm$0.44|
> ||GPT-5|36.03|35.07|35.93|36.62|35.96|35.66|36.23|36.16|35.96$\pm$0.42|

---

> ### Author Response · Authors · 2025-11-23
> **Rebuttal (2/3)**
>
> **[W2] One of the main contributions of this benchmark concerns subtask-level verification, which is important and useful for complex information retrieval tasks that is oft-neglected in many benchmarks. However, the paper fails to discuss this contribution in relation to previous work in the literature. For one, HotpotQA (which is cited by this paper) introduced supporting fact evaluation for reasoning, which is a proxy to checking the necessary reasoning steps are reached to arrive at the final answer. As a more recent example, the Agent Company [1] uses subtask rewards to evaluate agents on long-horizon agent tasks.**
>
> Thank you for your constructive comments. We have added a more detailed discussion of how HotpotQA and TheAgentCompany differ from our proposed VeriWeb benchmark as follows:
>
> (1) HotpotQA evaluates multi-hop question answering on a static Wikipedia corpus and provides sentence-level supporting fact annotations that indicate the evidence used to answer each final question.
> However, **the standard evaluation focuses on the correctness of the final answer and its associated supporting facts for a single global question**, without explicitly defining intermediate subgoals that can be evaluated as separate steps in a longer information-seeking workflow. In contrast, VeriWeb explicitly decomposes each realistic information-seeking task into a sequence of interdependent subtasks, each with its own sub-instruction and answer. Each subtask is designed to serve as a valid entry point, enabling agent evaluation at different stages of the overall task.
>
> (2) TheAgentCompany benchmarks long-horizon workplace tasks in a self-contained software company environment, where **checkpoint-based and partial credit evaluation is defined over environment state**, for example, whether particular files are created or fields are filled in internal web applications, which is well-suited to office process automation.
> In contrast, VeriWeb targets open web information seeking tasks and evaluates agents at the level of subtask goals rather than individual low-level actions, focusing on whether the objective of each subtask has been correctly achieved. This outcome-oriented, subtask-level evaluation provides a more informative supervision signal and supports open-ended interaction within each subtask, encouraging agents to explore diverse strategies to accomplish the subtask goal rather than adhering to a fixed action sequence.
>
> *These related works have been updated in Appendix E (Page 36) of the revised manuscript.*
>
> ---
>
> **[W3] Some claims in the paper are unsubstantiated. For instance, L360 states "Search engine agents, constrained to passive retrieval, typically achieve the lowest success rates." But the agents that achieved the lowest success rates in Table 2 are Browser-Use Agents and Multi-agent Systems.**
>
> Sorry for the confusion. This is a typo. We initially write the analysis when the search engine agent is not yet well-tuned. After we improve its performance, we forget to update this sentence accordingly. We thank the reviewer for pointing out this issue, and we have updated the corresponding analysis in the revised version.
>
> In addition, during the rebuttal period, we also include new results with the search engine agent using GPT-5. The results in Table R3 show that, although the search engine agent only uses a simple search tool, its performance is comparable to the other agents. This suggests that on our benchmark, the most important capability is still effective use of search, and that adding more complex tools does not necessarily bring additional gains. In some cases, the extra complexity may even make it harder for the model to plan effectively. *These new results have been updated in Section 4.2 (Page 7) of the revised manuscript.*
>
> Table R3 (Table 2 in the revision): Comparison of the search engine agent using OpenAI-o3 and GPT-5 on VeriWeb across five domains. Please refer to Table 2 in the revision for the results of all compared agents.
> |Model|Scientific||Finance||Technology||Arts||Social||Overall||
> |-|-:|-:|-:|-:|-:|-:|-:|-:|-:|-:|-:|-:|
> |     |SR (%)|CR (%)|SR (%)|CR (%)|SR (%)|CR (%)|SR (%)|CR (%)|SR (%)|CR (%)|SR (%)|CR (%)|
> |OpenAI-o3|0.0|17.6|0.0|18.2|7.4|28.0|13.5|37.6|3.1|20.6|6.0|26.1|
> |GPT-5|2.7 |21.1 |1.8 |16.3 |5.6 |30.4 |24.7 |51.8 |4.6 |28.3 |9.9 |32.5|

---

> ### Author Response · Authors · 2025-11-23
> **Rebuttal (3/3)**
>
> **[Q1] Why does the paper use Browser actions as the main metric for step / action efficiency? Just because this is how humans do it with a browser doesn't mean it's the most effective (as evidenced by the results) or efficient way for agents to do it.**
>
> Sorry for the confusion. As stated in the original manuscript, the action count quantifies the planning effectiveness of agents by measuring the number of steps required to arrive at the final answer. The definition of a "step" or "action" depends on the agent paradigm. Different agent paradigms have different action spaces, which are not the same as human actions. **We do not enforce a single human-style browser action space for this metric.** Therefore, **the action count between different agent paradigms or humans is not directly comparable**. *These clarifications have been updated in Section 4.1 (Page 6) of the revised manuscript.*
>
> In the "Analysis of Action Efficiency" section, we mainly compare the action counts of different models within the browser-use agents paradigm. This is because these open-source browser-use frameworks provide a rich and explicit set of tool calls and browser actions for LLM agents. Note that these **browser-use agents do not interact with the browser through human-style coordinate clicks on screenshots**. Instead, they parse the HTML DOM tree and operate on specific web elements. The agent itself decides which tools and actions to use to complete a task, so comparing action counts within this paradigm is meaningful. We also observe that a strong model like GPT-5 is more efficient than other baselines and can complete tasks with a very small number of actions.
>
> ---
>
> **[Q2] Some examples in Figure 4 have SR that's 0 < SR < 1, does this mean that SR is not a binary 0/1 metric, or is this averaged over several runs? What does it mean if a task is 50% successful?**
>
> Sorry for the confusion. In Figure 4 of the original manuscript, we report the distribution of task success rates under different agent paradigms. For each task, the success rate is the average over all models within that agent paradigm. For example, for the deep research SR, we average the success rates of four models within that paradigm: Gemini-2.5-Flash, Gemini-2.5-Pro, OpenAI-o4-mini, and OpenAI-o3. Therefore, if a task has a 50\% success rate, it means that only two of the four models succeed on that task. *These clarifications have been updated in Section 4.3 (Page 8) of the revised manuscript.*

---

> ### Author Response · Authors · 2025-11-27
> **Looking Forward to your Reevaluation**
>
> Dear Reviewer tn1J,
>
> We are glad that the reviewer appreciates our attempt, and sincerely thank you for the constructive comments. As suggested, we have additionally provided detailed clarifications for our subtask-level verification and action efficiency, and included further discussions and experiments on LLM-as-a-Judge agreement between human and LLM. Please let us know if you have other questions or comments.
>
> Since the discussion window has less than a week remaining, we sincerely look forward to your reevaluation of our work and would very appreciate it if you could raise your score to boost our chance of more exposure to the community. Thank you very much!
>
> Best regards,
>
> Authors of VeriWeb

---

> ### Comment · Reviewer_tn1J · 2025-11-27
> **Thank you for your response.**
>
> Thanks for the response and discussion.
>
> I appreciate the added human evaluation analysis, it helps better substantiate the reliability of the evaluation protocol.
>
> The response states
> >TheAgentCompany benchmarks long-horizon workplace tasks in a self-contained software company environment, where checkpoint-based and partial credit evaluation is defined over environment state, for example, whether particular files are created or fields are filled in internal web applications, which is well-suited to office process automation. In contrast, VeriWeb targets open web information seeking tasks and evaluates agents at the level of subtask goals rather than individual low-level actions, focusing on whether the objective of each subtask has been correctly achieved.
>
> I fail to see the distinction the authors are trying to make here. TheAgentCompany's checkpoint-based evaluation is also at the level of goals rather than individual low-level actions (it evaluates the **result** of those actions but not the actions themselves), and these can be viewed also as **subgoals** towards achieving the final task.

---

> > ### Author Response · Authors · 2025-11-28
> > **Thank you for your response!**
> >
> > Thank you for your response! We are glad that the reviewer appreciates our attempt and provides positive support, and we apologize for any confusion caused by our previous response. TheAgentCompany's checkpoint-based evaluation is indeed at the level of subtask goals rather than individual low-level actions. We would like to clarify that our setting differs from TheAgentCompany mainly in the following aspects:
> >
> > - **(1) Task type.** TheAgentCompany builds a **simulated software development company** environment with **internal websites and data**, where agents perform tasks such as software engineering, project management, and financial analysis. In contrast, VeriWeb targets **open-ended information seeking** in **realistic web environments**, focusing on long-horizon web tasks that require large-scale retrieval and synthesis of information from diverse real-world websites.
> >
> > - **(2) Subtask evaluation.** In TheAgentCompany, checkpoints are tightly coupled to the underlying environment state. As a result, it is **nontrivial to start the evaluation from an intermediate subtask**, because one must restore the entire environment to the corresponding state. In VeriWeb, each subtask can easily serve as an independent starting point, because the ground truth answers to all earlier subtasks are stored as text and can be given to the agent as contextual input.
> >
> > - **(3) Released artifact.** TheAgentCompany primarily provides a benchmark environment for evaluating LLM agents that act as digital workers, but without releasing related human trajectories. VeriWeb releases a **high-cost, human-annotated dataset** of 302 verifiable long-chain task trajectories across diverse real-world domains, capturing both long-chain complexity and subtask-level verifiability. On top of this dataset, we designed the VeriWeb benchmark, which supports fine-grained analysis of existing agent capabilities.
> >
> > We have updated these descriptions in Appendix E (page 36) of the revised manuscript. We again sincerely thank you for your time and constructive comments, which have helped us significantly improve the quality of the paper.

---

### Official Review · Reviewer_Vcuj · 2025-11-01

**Soundness:** 3
**Presentation:** 3
**Contribution:** 3
**Rating:** 6
**Confidence:** 4

**Summary:**

The authors introduce a benchmark that tests web-based language agents on complex, multi-hop information-seeking tasks. Each task comes from a human browsing session, where annotators solved real web problems and recorded the steps needed to find and verify facts. The dataset breaks these trajectories into verifiable subtasks. VeriWeb scores agents using three measures: Success Rate for full-task completion, Completion Rate for subtask accuracy, and Action Efficiency for how effectively they use the browser. The authors test several large language models and find that agents often locate some facts correctly but struggle to plan, search deeply, and keep results consistent across steps.

**Strengths:**

- Tasks are long-chain and information-dense, combining multi-hop retrieval and synthesis with subtask-level verifiability.
- The benchmark introduces several evaluation metrics, including task success rate, completion rate, and action efficiency.
- Human-annotated trajectories provide empirically grounded task structures.

**Weaknesses:**

- The benchmark’s subtask-level verifiability requires each sub-answer to be fixed and unambiguous, but real-world web tasks often involve context-dependent or time-sensitive information. This design choice may therefore underrepresent the uncertainty present in realistic settings.
- The absence of a human performance baseline makes it hard to interpret how well current agents perform relative to human proficiency on the same tasks.
- Evaluation only uses gpt-4o as the judge, with no analysis on potential LLM judge bias or comparison against human-annotated labels.

**Questions:**

- Would it be possible to run a quantitative breakdown of failure modes across agents?

---

> ### Author Response · Authors · 2025-11-23
> **Rebuttal (1/4)**
>
> We sincerely appreciate the reviewer for highlighting the strengths of VeriWeb, including `the long-chain and subtask-level design`, `the benchmark with several evaluation metrics`, and `the empirically grounded human-annotated trajectories`. We have carefully revised the manuscript according to your valuable suggestions. Below we address the main points raised in the review.
>
> ---
>
> **[W1] The benchmark’s subtask-level verifiability requires each sub-answer to be fixed and unambiguous, but real-world web tasks often involve context-dependent or time-sensitive information. This design choice may therefore underrepresent the uncertainty present in realistic settings.**
>
> Thank you for the valuable comment. (1) **The tradeoff between verifiability and uncertainty is an interesting issue, and is indeed central to our design.** Existing information-seeking benchmarks largely fall into two categories. The first targets single-fact retrieval, which is fully **verifiable** but falls short of capturing realistic tasks that require information-rich, synthesized outputs. The second targets deep research report-style generation, which captures open-ended **uncertainty** but makes direct ground-truth evaluation difficult, and often relies on rubric-based scoring for subjective aspects such as coherence and completeness. In contrast, our primary goal with VeriWeb is to benchmark one core capability that remains challenging in realistic web environments: **large-scale retrieval and synthesis of information from diverse sources (similar to deep research) that still yields a verifiable answer (similar to single-fact retrieval)**.
>
> (2) It is also important to clarify that **subtask-level verifiability does not eliminate uncertainty in how agents solve each subtask**. While every subtask has a fixed answer, the path to that answer is deliberately open-ended. Agents may pursue different information-gathering strategies, such as consulting an official government portal or a well-maintained wiki entry when collecting economic data. All of these choices reflect realistic variation in real-world workflows. Because the benchmark does not prescribe any specific action sequence, **agents must still decide which sources to trust, manage conflicting evidence across pages, and maintain coherent context throughout long navigation chains**. These factors introduce meaningful uncertainty even under a verifiable subtask structure, enabling stable evaluation while preserving the complexity of genuine information-seeking behavior.
>
> We believe this tradeoff is reasonable as a first step toward verifiable long-chain web benchmarks.

---

> ### Author Response · Authors · 2025-11-23
> **Rebuttal (2/4)**
>
> **[W2] The absence of a human performance baseline makes it hard to interpret how well current agents perform relative to human proficiency on the same tasks.**
>
> Thank you for the insightful comment. In the VeriWeb benchmark, we regard human performance as essentially at ceiling (very close to perfect), because all ground truth answers in our dataset are indeed given by human annotators.
>
> (1) As discussed in our paper, our benchmark targets large-scale retrieval and synthesis of information from diverse sources. These are exactly the kinds of tasks where **current agents still underperform, yet humans routinely solve them in everyday and professional settings**. Therefore, to construct a reliable reference, all task answers are produced by human experts rather than models.
>
> (2) We recruited 30 human annotators, all with at least a bachelor's degree, at a cost of \\$50 per annotation, to provide the answers that serve as the gold standard ground truth. Each annotation session typically lasts 1 to 2 hours. After the initial annotation, two review teams (each with 2 reviewers), also with at least a bachelor's degree, at a cost of \\$20 per review, independently check whether each sample satisfies the task requirements. Samples that do not fully meet the requirements are sent back for revision and then re-checked, and in practice, most items are approved after no more than two rounds of revision. Given this process, **we expect human performance on these tasks to be very close to 100\%**.
>
> (3) To more directly address the concern about a human performance baseline, **we have additionally designed an experiment to measure human performance under a time limit** (12 minutes per task, which roughly matches the typical completion time of the deep research agents and browser-use agents). Based on the task difficulty levels in our original experiment, we randomly sampled 10 tasks from each difficulty level and recruited 5 additional human annotators (all with at least a bachelor’s degree, at a cost of \\$20 per annotation) to complete them under this time limit. In this setting, humans did not successfully complete any tasks, so we report only task completion rates. The results in Table R1 show that **human performance drops substantially under this constraint**. This illustrates a regime where agents can surpass time-limited human performance, even though the ground truth remains defined by unconstrained human experts.
>
> Table R1 (Table 9 in the revision): Comparison of task completion rates on VeriWeb. Human performance is measured under a 12-minute time limit per task. Deep research agents typically require about 12 minutes per task, search engine agents about 4 minutes per task, browser-use agents about 10 minutes per task, and multi-agent systems about 40 minutes per task.
> |Model|Level 1|Level 2|Level 3|Level 4|Level 5|
> |-|-:|-:|-:|-:|-:|
> |**Deep Research Agents**||||||
> |Gemini-2.5-Flash|93.0|47.0|24.0|1.0|0.0|
> |Gemini-2.5-Pro|84.0|56.0|8.0|6.0|0.0|
> |OpenAI-o4-mini|63.0|34.0|13.0|3.0|0.0|
> |OpenAI-o3|75.0|63.0|8.0|7.0|0.0|
> |Search Engine Agents||||||
> |OpenAI-o3|46.0|42.0|21.0|5.0|0.0|
> |GPT-5|80.0|58.0|0.0|6.0|0.0|
> |**Browser-Use Agents**||||||
> |Qwen-VL-Max|17.0|12.0|0.0|0.0|0.0|
> |DeepSeek-V3.1|23.0|38.0|4.0|4.0|0.0|
> |Gemini-2.5-Flash|40.0|41.0|7.0|1.0|1.0|
> |Gemini-2.5-Pro|45.0|35.0|16.0|4.0|4.0|
> |Claude-3.7-Sonnet|78.0|31.0|16.0|5.0|3.0|
> |Claude-4.0-Sonnet|42.0|16.0|4.0|2.0|0.0|
> |OpenAI-o3|62.0|47.0|5.0|5.0|2.0|
> |GPT-4o|18.0|15.0|2.0|0.0|0.0|
> |GPT-4.1|57.0|35.0|10.0|3.0|4.0|
> |GPT-5|34.0|35.0|5.0|3.0|0.0|
> |**Multi-Agent Systems**||||||
> |Qwen3-235B|17.0|14.0|1.0|0.0|0.0|
> |DeepSeek-V3.1|43.0|6.0|0.0|0.0|0.0|
> |OpenAI-o3|69.0|62.0|6.0|4.0|0.0|
> |GPT-5|17.0|15.0|9.0|0.0|0.0|
> |**Human**|47.0|40.0|15.0|6.0|1.0|
>
> *These clarifications have been updated in Appendix D.2 (Page 27--28) of the revised manuscript.*

---

> ### Author Response · Authors · 2025-11-23
> **Rebuttal (3/4)**
>
> **[W3] Evaluation only uses gpt-4o as the judge, with no analysis on potential LLM judge bias or comparison against human-annotated labels.**
>
> Thanks for your insightful comment. As suggested, we have additionally conducted experiments comparing human and LLM judgment (using different LLMs and multiple evaluation runs) in order to validate the stability and reliability of our evaluation protocol.
>
> **(1) Human judgment.** Since the deep research agent using OpenAI-o3 performs well in the original experiment, we use its responses for this analysis. In the original experiment, we use LLM as the judge and assign a score in the range $[0, 10]$ to each response. Here, we partition these scores into five intervals, and randomly sample 10 responses from each interval, resulting in 50 responses. Human annotators then rescore these responses, and we compare the human scores with the LLM scores.
>
> Table R2 reports the agreement matrix between LLM and human buckets. For example, the 20\% in the first row and second column means that among responses that LLM scores in $[0,2)$, 20\% receive a human score in $[2,4)$. The results show that **for low-quality and high-quality responses, human annotators and the LLM judge agree very well**, while **for medium-quality responses, the LLM judge tends to be more stringent and humans are slightly more lenient, assigning somewhat higher scores**. Overall, the trend is consistent, and both judges rank clearly good and clearly bad responses similarly. For all 50 samples, the overall Pearson correlation between LLM scores and human scores is 0.9195, and 92% of responses satisfy | LLM Score - Human Score | $\leq$ 2. These statistics indicate strong linear agreement between LLM and human judgments and show that most per-response disagreements are small in magnitude, which supports the use of the LLM judge on this dataset. *These new results have been updated in Section 4.5 (Page 10) of the revised manuscript.*
>
> Table R2 (Table 6 in the revision): Bucket-level agreement between human and LLM judgment (rows: LLM buckets, columns: human buckets).
> ||[0,2)|[2,4)|[4,6)|[6,8)|[8,10]|
> |---|---|---|---|---|---|
> |**[0,2)**|80\%|20\%|0\%|0\%|0\%|
> |**[2,4)**|0\%|20\%|70\%|10\%|0\%|
> |**[4,6)**|0\%|0\%|30\%|70\%|0\%|
> |**[6,8)**|0\%|0\%|0\%|10\%|90\%|
> |**[8,10]**|0\%|0\%|0\%|0\%|100\%|
>
> **(2) LLM judgment using different LLMs and multiple evaluation runs.** In the original experiments, we use OpenAI-o3 as the judge. Here, we use GPT-5 as another judge and repeat each evaluation eight times per judge to measure both cross-judge agreement and stability under resampling. Due to time constraints, we conduct this analysis on the deep research agent with different backbone models.
>
> The results in Table R3 show that (1) For most models, the average scores under OpenAI-o3 and GPT-5 differ by at most about 2\%, showing **high consistency across judges**. The only exception is OpenAI-o4-mini, where GPT-5 assigns a lower average score by about 5\%. Manual inspection suggests that GPT-5 is particularly conservative on OpenAI-o4-mini outputs that contain the correct answer mixed with irrelevant or noisy content, and often treats such responses as incorrect. Importantly, regardless of which judge we use, the ranking of models remains stable and always satisfies OpenAI-o3 > Gemini-2.5-Pro > Gemini-2.5-Flash > OpenAI-o4-mini. (2) For each judge, the standard deviation across 8 runs is within about 1\%, which shows that **repeated sampling of the LLM judge yields very similar aggregate scores and that the evaluation is stable**. *These new results have been updated in Appendix D.1 (Page 27) of the revised manuscript.*
>
> Table R3 (Table 8 in the revision): Comparison of the task completion rate of deep research agents across different LLM judges and multiple evaluation runs.
> |Model|Judge|Run 1|Run 2|Run 3|Run 4|Run 5|Run 6|Run 7|Run 8|Average|
> |-|-|-|-|-|-|-|-|-|-|-|
> |Gemini-2.5-Flash|OpenAI-o3|31.16|30.17|30.79|30.93|30.23|31.03|30.63|30.79|30.72$\pm$0.33|
> ||GPT-5|29.40|29.14|29.07|29.64|29.17|29.54|29.67|29.54|29.40$\pm$0.22|
> |Gemini-2.5-Pro|OpenAI-o3|33.94|33.97|34.34|34.30|34.07|34.14|35.26|33.54|34.20$\pm$0.47|
> ||GPT-5|32.78|32.95|33.11|33.64|33.25|33.31|33.11|33.28|33.18$\pm$0.24|
> |OpenAI-o4-mini|OpenAI-o3|25.93|25.89|26.03|25.86|25.86|25.99|26.03|26.09|25.96$\pm$0.08|
> ||GPT-5|21.23|20.03|20.43|20.96|20.60|22.05|21.72|20.99|21.00$\pm$0.62|
> |OpenAI-o3|OpenAI-o3|36.26|36.03|37.19|36.72|36.13|35.86|35.79|36.23|36.28$\pm$0.44|
> ||GPT-5|36.03|35.07|35.93|36.62|35.96|35.66|36.23|36.16|35.96$\pm$0.42|

---

> ### Author Response · Authors · 2025-11-23
> **Rebuttal (4/4)**
>
> **[Q1] Would it be possible to run a quantitative breakdown of failure modes across agents?**
>
> Thanks for your valuable suggestion. We have additionally conducted a quantitative breakdown of different failure modes using GPT-5 as a judge, including (a) misinformation, (b) incomplete result, (c) retrieval failure, and (d) irrelevant result. The detailed evaluation prompt is provided in Appendix C. Note that a single response may belong to several failure modes at the same time.
>
> The results in Table R2 show that misinformation and incomplete results are the two dominant failure modes across all agent paradigms, while retrieval failures and irrelevant results occur less frequently but are still non-negligible. Deep research and search engine agents tend to fail primarily through misinformation, suggesting that they often retrieve relevant evidence but hallucinate or approximate key numerical or factual details. In contrast, the browser-use agents and multi-agent systems are more frequently dominated by incomplete results, indicating that these agents are better at locating relevant sources but struggle to aggregate and exhaustively cover all required items in long-chain tasks. Across all settings, retrieval failures rarely account for the majority of errors, highlighting that the central bottleneck lies in accurate synthesis and coverage rather than merely finding at least one relevant document. *These new results have been updated in Section 4.4 (Page 10) of the revised manuscript.*
>
> Table R4: Comparison of different failure modes on VeriWeb.
> |Model|Misinformation|Incomplete Result|Retrieval Failure|Irrelevant Result|
> |-|-|-|-|-|
> |**Deep Research Agent**||||
> |Gemini-2.5-Flash|44\% (254)|34\% (197)|14\% (82)|8\% (42)|
> |Gemini-2.5-Pro|44\% (213)|36\% (172)|11\% (54)|9\% (38)|
> |OpenAI-o4-mini|50\% (249)|36\% (178)|9\% (46)|5\% (20)|
> |OpenAI-o3|63\% (248)|30\% (118)|3\% (12)|4\% (11)|
> |**Search Engine Agents**||||
> |OpenAI-o3|62\% (226)|31\% (113)|4\% (16)|3\% (9)|
> |GPT-5|46\% (172)|42\% (158)|6\% (23)|6\% (16)|
> |**Browser-Use Agents**||||
> |Qwen-VL-Max|12\% (51)|67\% (274)|14\% (58)|7\% (24)|
> |DeepSeek-V3.1|21\% (100)|55\% (262)|17\% (81)|7\% (27)|
> |Gemini-2.5-Flash|25\% (125)|53\% (258)|12\% (58)|10\% (40)|
> |Gemini-2.5-Pro|34\% (149)|49\% (214)|12\% (53)|5\% (19)|
> |Claude-3.7-Sonnet|34\% (189)|38\% (212)|19\% (105)|9\% (40)|
> |Claude-4.0-Sonnet|19\% (85)|58\% (253)|16\% (73)|7\% (24)|
> |OpenAI-o3|50\% (183)|39\% (144)|5\% (20)|6\% (16)|
> |GPT-4o|22\% (125)|47\% (269)|24\% (140)|7\% (34)|
> |GPT-4.1|33\% (216)|37\% (246)|24\% (158)|6\% (34)|
> |GPT-5|17\% (81)|55\% (254)|20\% (94)|8\% (28)|
> |**Multi-Agent Systems**||||
> |Qwen3-235B|23\% (97)|56\% (239)|12\% (53)|9\% (31)|
> |DeepSeek-V3.1|12\% (55)|61\% (270)|17\% (76)|10\% (36)|
> |OpenAI-o3|42\% (168)|41\% (163)|8\% (32)|9\% (32)|
> |GPT-5|19\% (79)|60\% (243)|13\% (56)|8\% (25)|

---

> ### Author Response · Authors · 2025-11-27
> **Looking Forward to your Reevaluation**
>
> Dear Reviewer Vcuj,
>
> We are glad that the reviewer appreciates our attempt, and sincerely thank you for the constructive comments. As suggested, we have additionally provided detailed clarifications for the tradeoff between verifiability and uncertainty, and included further discussions and experiments on human evaluation and failure modes. Please let us know if you have other questions or comments.
>
> Since the discussion window has less than a week remaining, we sincerely look forward to your reevaluation of our work and would very appreciate it if you could raise your score to boost our chance of more exposure to the community. Thank you very much!
>
> Best regards,
>
> Authors of VeriWeb

---

### Official Review · Reviewer_Q1hS · 2025-11-01

**Soundness:** 3
**Presentation:** 3
**Contribution:** 3
**Rating:** 4
**Confidence:** 4

**Summary:**

The paper introduces VeriWeb, a new benchmark for evaluating long-horizon, web-based information-seeking agents. VeriWeb focuses on long-chain complexity (requiring multi-hop reasoning and synthesis across diverse sources) and subtask-level verifiability (allowing fine-grained evaluation of intermediate steps). The dataset comprises 302 human-annotated tasks across five domains, each decomposed into verifiable subtasks. Experiments with multiple agents powered by different foundation models show low success rates, highlighting the difficulty of realistic web reasoning.

**Strengths:**

- Proposes a novel benchmark emphasizing both long-chain reasoning and verifiable subtasks.

- The dataset is diverse and human-annotated, covering five realistic domains.

- The experimental evaluation is comprehensive, testing multiple agent paradigms and models.

- The paper provides insightful analyses of action efficiency and task difficulty, helping identify weaknesses in current web agents.

**Weaknesses:**

- Unclear which LLM generated task instructions and subtasks.

- The reasonableness of subtask decomposition is not independently validated.

- Details of the human demonstration process (e.g., annotator number, quality checks, or fairness) are limited.

- Many tasks involve hundreds of steps, but efficiency guarantees or annotation consistency are not analyzed.

- The LLM-as-a-Judge metric may not align with human evaluation; human verification would strengthen credibility.

- Single-run experiments due to API costs limit statistical reliability.

- No human baseline is reported to contextualize task difficulty.

- Error analysis could be broader and more quantitative.

**Questions:**

- Which model was used to generate and decompose the tasks?

- How is the quality or coherence of subtask decomposition verified?

- What measures ensure fairness and accuracy in human demonstrations?

- How do the authors justify using LLM-as-a-Judge without human correlation studies?

- Could results be re-evaluated with multiple runs perhaps with open-source models to report variance?

- Is there a plan to report human performance per difficulty level?

- Can the authors share the API cost estimates and efficiency trade-offs?

---

> ### Author Response · Authors · 2025-11-23
> **Rebuttal (1/4)**
>
> We sincerely appreciate the reviewer for highlighting the strengths of VeriWeb, including the `novel benchmark`, the `diverse human-annotated dataset`, and the `comprehensive experimental evaluation` with the `insightful analyses`. We have carefully revised the manuscript according to your valuable suggestions. Below we address the main points raised in the review.
>
> ---
>
> **[W1 & Q1] Unclear which LLM generated task instructions and subtasks. Which model was used to generate and decompose the tasks?**
>
> Sorry for the confusion. In our early experiments, we primarily used GPT-4.1. After GPT-5 was released, we switched to GPT-5 for generating and decomposing the tasks. *These clarifications have been updated in Appendix A (Page 16) of the revised manuscript.*
>
> ---
>
> **[W2 & Q2] The reasonableness of subtask decomposition is not independently validated. How is the quality or coherence of subtask decomposition verified?**
>
> Thanks for the valuable comment. In our data construction pipeline, the reasonableness of subtask decomposition is explicitly checked in the model-based validation stage. We use a strong LLM (early GPT-4.1 and later GPT-5) as an automatic reviewer and instruct it via a detailed prompt to evaluate each candidate instruction with respect to several criteria, including (1) Structure and format compliance, (2) Task difficulty and long-chain nature, (3) Verifiability and uniqueness, (4) Feasibility, (5) Innovation and diversity, and (6) Reasonableness of scope limitations. The detailed validation prompt is provided in Appendix A.2.
>
> For each candidate, the review model outputs one of three labels:
> - **"Pass":** the instruction is accepted and moves to the next round or to the final pool.
> - **"Revise":** the instruction is useful in principle but has concrete, fixable issues. These cases are returned to the generation phase with feedback, revised, and then re-submitted to the review model.
> - **"Reject":** the instruction is fundamentally flawed or not worth revising and is discarded.
>
> We run 3 model-based validation rounds for each instance. The motivation is to reduce the probability that a low-quality instance passes due to a single stochastic error of the reviewer model. Only instances that are consistently labeled as "Pass" across all model-based validation rounds survive this stage. *These clarifications have been updated in Appendix A.2 (Page 17--20) of the revised manuscript.*

---

> ### Author Response · Authors · 2025-11-23
> **Rebuttal (2/4)**
>
> **[W3 & W4 & Q3] Details of the human demonstration process (e.g., annotator number, quality checks, or fairness) are limited. Many tasks involve hundreds of steps, but efficiency guarantees or annotation consistency are not analyzed. What measures ensure fairness and accuracy in human demonstrations?**
>
> Sorry for the confusion. We partnered with a professional data annotation company that follows standardized procedures and internal quality audits. Specifically, our human demonstration collection follows a four-stage adversarial annotation protocol:
>
> - **Stage 1: Initial annotation.** We recruited **30 human annotators**, all with at least a bachelor's degree, at a cost of \\$50 per annotation. Human annotators complete the task following a standardized data collection document (see Appendix B.2 for details). They conduct keyword-based searches on credible sources, record the entire retrieval process step by step without using any AI tools, and follow requirements on evidence highlighting, screen recording quality, and structured summarization of subtask-level answers. **Each annotation session generally lasts 1 to 2 hours**. These initial trajectories form the base demonstrations.
> - **Stage 2: Adversarial dual review.** Each initial trajectory is then examined by **2 independent review teams**, with 2 reviewers per team and all reviewers holding at least a bachelor’s degree. Reviewers also follow a standardized data review document (see Appendix B.3 for details) that covers search logic, evidence consistency, accuracy checks, and compliance with recording constraints. They verify that each retrieval step matches the corresponding sub-instruction, that recorded pages support the submitted answers, and that no prohibited operations occur. **The two teams do not see each other's comments**, which creates an adversarial, multi-perspective inspection of long, multi-step trajectories that is designed to stress test the process rather than only verify the final answer.
> - **Stage 3: Iterative refinement.** The original annotator receives consolidated feedback from both review teams and revises the trajectory accordingly. If the trajectory does not fully satisfy the requirements, it is returned for further revision and rechecked. In practice, **most items are accepted after no more than 2 refinement rounds**. This iterative loop, combined with shared guidelines, helps align annotators on step granularity and reasoning style, which in turn improves cross-sample consistency for these long trajectories.
> - **Stage 4: Final verification.** A separate reviewer, who did not participate in the earlier stages, **performs a final pass** before the example is admitted into the released dataset. This reviewer validates correctness, compliance with all document requirements, internal consistency between recorded evidence and submitted answers, and the absence of problematic shortcuts or unfair treatment. Items that fail any requirement are rejected.
>
> Moreover, we employ a **game-theoretic incentive mechanism** to encourage deep and careful review. The two review teams share a fixed bonus pool (\\$20 per review) with a high variance asymmetric allocation: 80\% of the bonus is awarded to the team whose feedback is judged more comprehensive, actionable, and insightful, and 20\% to the other team. This design creates strong incentives for reviewers to maximize review depth and precision instead of free riding, and it rewards the identification of subtle errors, omissions, or potential sources of bias.
>
> Overall, this pipeline (1) encourages exhaustive error detection through adversarial, multi-team scrutiny, (2) mitigates individual annotator bias via independent reviews and a separate final verifier, and (3) enforces consistency through standardized protocols, iterative refinement, and incentives aligned with review quality. **These measures are specifically designed to provide the fairness, accuracy, and reliability guarantees needed for demonstrations that span hundreds of retrieval and reasoning steps.** *These clarifications have been updated in Appendix B (Page 20) of the revised manuscript.*

---

> ### Author Response · Authors · 2025-11-23
> **Rebuttal (3/4)**
>
> **[W5 & Q4 & W8 & Q5] The LLM-as-a-Judge metric may not align with human evaluation; human verification would strengthen credibility. How do the authors justify using LLM-as-a-Judge without human correlation studies? Error analysis could be broader and more quantitative. Could results be re-evaluated with multiple runs perhaps with open-source models to report variance?**
>
> Thanks for your insightful comment. As suggested, we have additionally conducted experiments comparing human and LLM judgment (using different LLMs and multiple evaluation runs) in order to validate the stability and reliability of our evaluation protocol.
>
> **(1) Human judgment.** Since the deep research agent using OpenAI-o3 performs well in the original experiment, we use its responses for this analysis. In the original experiment, we use LLM as the judge and assign a score in the range $[0, 10]$ to each response. Here, we partition these scores into five intervals, and randomly sample 10 responses from each interval, resulting in 50 responses. Human annotators then rescore these responses, and we compare the human scores with the LLM scores.
>
> Table R1 reports the agreement matrix between LLM and human buckets. For example, the 20\% in the first row and second column means that among responses that LLM scores in $[0,2)$, 20\% receive a human score in $[2,4)$. The results show that **for low-quality and high-quality responses, human annotators and the LLM judge agree very well**, while **for medium-quality responses, the LLM judge tends to be more stringent and humans are slightly more lenient, assigning somewhat higher scores**. Overall, the trend is consistent, and both judges rank clearly good and clearly bad responses similarly. For all 50 samples, the overall Pearson correlation between LLM scores and human scores is 0.9195, and 92% of responses satisfy | LLM Score - Human Score | $\leq$ 2. These statistics indicate strong linear agreement between LLM and human judgments and show that most per-response disagreements are small in magnitude, which supports the use of the LLM judge on this dataset. *These new results have been updated in Section 4.5 (Page 10) of the revised manuscript.*
>
> Table R1 (Table 6 in the revision): Bucket-level agreement between human and LLM judgment (rows: LLM buckets, columns: human buckets).
> ||[0,2)|[2,4)|[4,6)|[6,8)|[8,10]|
> |---|---|---|---|---|---|
> |**[0,2)**|80\%|20\%|0\%|0\%|0\%|
> |**[2,4)**|0\%|20\%|70\%|10\%|0\%|
> |**[4,6)**|0\%|0\%|30\%|70\%|0\%|
> |**[6,8)**|0\%|0\%|0\%|10\%|90\%|
> |**[8,10]**|0\%|0\%|0\%|0\%|100\%|
>
> **(2) LLM judgment using different LLMs and multiple evaluation runs.** In the original experiments, we use OpenAI-o3 as the judge. Here, we use GPT-5 as another judge and repeat each evaluation eight times per judge to measure both cross-judge agreement and stability under resampling. Due to time constraints, we conduct this analysis on the deep research agent with different backbone models.
>
> The results in Table R2 show that (1) For most models, the average scores under OpenAI-o3 and GPT-5 differ by at most about 2\%, showing **high consistency across judges**. The only exception is OpenAI-o4-mini, where GPT-5 assigns a lower average score by about 5\%. Manual inspection suggests that GPT-5 is particularly conservative on OpenAI-o4-mini outputs that contain the correct answer mixed with irrelevant or noisy content, and often treats such responses as incorrect. Importantly, regardless of which judge we use, the ranking of models remains stable and always satisfies OpenAI-o3 > Gemini-2.5-Pro > Gemini-2.5-Flash > OpenAI-o4-mini. (2) For each judge, the standard deviation across 8 runs is within about 1\%, which shows that **repeated sampling of the LLM judge yields very similar aggregate scores and that the evaluation is stable**. *These new results have been updated in Appendix D.1 (Page 27) of the revised manuscript.*
>
> Table R2 (Table 8 in the revision): Comparison of the task completion rate of deep research agents across different LLM judges and multiple evaluation runs.
> |Model|Judge|Run 1|Run 2|Run 3|Run 4|Run 5|Run 6|Run 7|Run 8|Average|
> |-|-|-|-|-|-|-|-|-|-|-|
> |Gemini-2.5-Flash|OpenAI-o3|31.16|30.17|30.79|30.93|30.23|31.03|30.63|30.79|30.72$\pm$0.33|
> ||GPT-5|29.40|29.14|29.07|29.64|29.17|29.54|29.67|29.54|29.40$\pm$0.22|
> |Gemini-2.5-Pro|OpenAI-o3|33.94|33.97|34.34|34.30|34.07|34.14|35.26|33.54|34.20$\pm$0.47|
> ||GPT-5|32.78|32.95|33.11|33.64|33.25|33.31|33.11|33.28|33.18$\pm$0.24|
> |OpenAI-o4-mini|OpenAI-o3|25.93|25.89|26.03|25.86|25.86|25.99|26.03|26.09|25.96$\pm$0.08|
> ||GPT-5|21.23|20.03|20.43|20.96|20.60|22.05|21.72|20.99|21.00$\pm$0.62|
> |OpenAI-o3|OpenAI-o3|36.26|36.03|37.19|36.72|36.13|35.86|35.79|36.23|36.28$\pm$0.44|
> ||GPT-5|36.03|35.07|35.93|36.62|35.96|35.66|36.23|36.16|35.96$\pm$0.42|

---

> ### Author Response · Authors · 2025-11-23
> **Rebuttal (4/4)**
>
> **[W6 & Q7] Single-run experiments due to API costs limit statistical reliability. Can the authors share the API cost estimates and efficiency trade-offs?**
>
> Thank you for raising this point. As discussed in our paper, our benchmark is designed for large-scale retrieval and synthesis of information from diverse sources. These information-seeking tasks are **extremely expensive in terms of both time and tokens**. In our setup, deep research agents typically require about 12 minutes per task, search engine agents about 4 minutes per task, browser-use agents about 10 minutes per task, and multi-agent systems about 40 minutes per task. Moreover, all the experiments for this paper incurred about \\$5,194.40 in API charges. Among the configurations we used, the deep research agent using OpenAI-o3 was the most expensive, at roughly \\$2 per sample, so **running 302 samples costs  ~\\$600**. In contrast, a deep research agent using OpenAI-o4-mini costs roughly \\$0.1 per sample, so 302 samples cost ~\\$30.
> Given the number of tasks in our benchmark, the resulting wall-clock time and API usage substantially exceeded our initial expectations, which is why we were not able to afford multiple independent runs for each setting.
>
> ---
>
> **[W7 & Q6] No human baseline is reported to contextualize task difficulty. Is there a plan to report human performance per difficulty level?**
>
> Thank you for the insightful comment. In the VeriWeb benchmark, we regard human performance as essentially at ceiling (very close to perfect), because all ground truth answers in our dataset are indeed given by human annotators.
>
> (1) As discussed in our paper, our benchmark targets large-scale retrieval and synthesis of information from diverse sources. These are exactly the kinds of tasks where **current agents still underperform, yet humans routinely solve them in everyday and professional settings**. Therefore, to construct a reliable reference, all task answers are produced by human experts rather than models.
>
> (2) As stated in our response to [W3 & W4 & Q3], we recruited human annotators to provide the answers that serve as the gold standard ground truth. Each annotation session typically lasts 1 to 2 hours. After the initial annotation, the internal review teams will check whether each sample satisfies the task requirements. Samples that do not fully meet the requirements are sent back for revision and then re-checked, and in practice, most items are approved after no more than two rounds of revision. Given this process, **we expect human performance on these tasks to be very close to 100\%**.
>
> (3) To more directly address the concern about a human performance baseline, **we have additionally designed an experiment to measure human performance under a time limit** (12 minutes per task, which roughly matches the typical completion time of the deep research agents and browser-use agents). Based on the task difficulty levels in our original experiment, we randomly sampled 10 tasks from each difficulty level and recruited 5 additional human annotators (all with at least a bachelor’s degree, at a cost of \\$20 per annotation) to complete them under this time limit. In this setting, humans did not successfully complete any tasks, so we report only task completion rates. The results in Table R3 show that **human performance drops substantially under this constraint**. This illustrates a regime where agents can surpass time-limited human performance, even though the ground truth remains defined by unconstrained human experts.
>
> Table R3 (Table 9 in the revision): Comparison of task completion rates on VeriWeb. Human performance is measured under a 12-minute time limit per task.
> |Model|Level 1|Level 2|Level 3|Level 4|Level 5|
> |-|-:|-:|-:|-:|-:|
> |**Deep Research Agents**||||||
> |Gemini-2.5-Flash|93.0|47.0|24.0|1.0|0.0|
> |Gemini-2.5-Pro|84.0|56.0|8.0|6.0|0.0|
> |OpenAI-o4-mini|63.0|34.0|13.0|3.0|0.0|
> |OpenAI-o3|75.0|63.0|8.0|7.0|0.0|
> |Search Engine Agents||||||
> |OpenAI-o3|46.0|42.0|21.0|5.0|0.0|
> |GPT-5|80.0|58.0|0.0|6.0|0.0|
> |**Browser-Use Agents**||||||
> |Qwen-VL-Max|17.0|12.0|0.0|0.0|0.0|
> |DeepSeek-V3.1|23.0|38.0|4.0|4.0|0.0|
> |Gemini-2.5-Flash|40.0|41.0|7.0|1.0|1.0|
> |Gemini-2.5-Pro|45.0|35.0|16.0|4.0|4.0|
> |Claude-3.7-Sonnet|78.0|31.0|16.0|5.0|3.0|
> |Claude-4.0-Sonnet|42.0|16.0|4.0|2.0|0.0|
> |OpenAI-o3|62.0|47.0|5.0|5.0|2.0|
> |GPT-4o|18.0|15.0|2.0|0.0|0.0|
> |GPT-4.1|57.0|35.0|10.0|3.0|4.0|
> |GPT-5|34.0|35.0|5.0|3.0|0.0|
> |**Multi-Agent Systems**||||||
> |Qwen3-235B|17.0|14.0|1.0|0.0|0.0|
> |DeepSeek-V3.1|43.0|6.0|0.0|0.0|0.0|
> |OpenAI-o3|69.0|62.0|6.0|4.0|0.0|
> |GPT-5|17.0|15.0|9.0|0.0|0.0|
> |**Human**|47.0|40.0|15.0|6.0|1.0|
>
> *These clarifications have been updated in Appendix D.2 (Page 27--28) of the revised manuscript.*

---

> ### Author Response · Authors · 2025-11-27
> **Looking Forward to your Reevaluation**
>
> Dear Reviewer Q1hS,
>
> We are glad that the reviewer appreciates our attempt, and sincerely thank you for the constructive comments. As suggested, we have additionally provided detailed clarifications for our human demonstration collection, and included further discussions and experiments on LLM-as-a-Judge agreement and human performance. Please let us know if you have other questions or comments.
>
> Since the discussion window has less than a week remaining, we sincerely look forward to your reevaluation of our work and would very appreciate it if you could raise your score to boost our chance of more exposure to the community. Thank you very much!
>
> Best regards,
>
> Authors of VeriWeb

---

### Official Review · Reviewer_76ET · 2025-11-01

**Soundness:** 2
**Presentation:** 3
**Contribution:** 2
**Rating:** 4
**Confidence:** 3

**Summary:**

This paper introduces VeriWeb, a novel verifiable long-chain benchmark, intended to address critical limitations in existing web agent evaluation—specifically the overreliance on single-fact retrieval and outcome-only validation. VeriWeb comprises 302 human-annotated tasks across five real-world domains. The benchmark mandates long-chain complexity (requiring large-scale retrieval, multi-hop reasoning, and information synthesis) and incorporates subtask-level verifiability. Experiments confirm the benchmark’s difficulty, showing that state-of-the-art LLM-powered agents achieve consistently poor performance on these complex, long-horizon tasks.

**Strengths:**

The primary contribution is the development of a benchmark that rigorously enforces two previously neglected dimensions: long-chain complexity (integrating breadth- and depth-oriented search) and subtask-level verifiability. This fine-grained decomposition is essential, providing an informative supervision signal and allowing for error localization, which outcome-only evaluation protocols fail to capture. The dataset, curated through a costly human-annotation process across diverse real-world domains, effectively serves its purpose by revealing significant performance gaps and underscoring current agent limitations in synthesis and complex retrieval.

**Weaknesses:**

This is a paper proposing a new web agent benchmark. However, a new benchmark must clearly state the problem it solves and rigorously demonstrate why this problem and the evaluation method are important, with the analysis of failure cases being able to guide the direction of field development.
The problem this paper addresses is relatively clear, and the proposal of the dataset and its construction method also have value. However, there is no particularly detailed justification for why evaluation should be conducted through subtasks (and other agent evaluation papers have proposed similar evaluation methods and metrics). Furthermore, the analysis of failure cases lacks sufficient depth and does not offer unique insights.
If this paper merely defines a new benchmark data generation process (with missing details on the data synthesis process) and conducts a certain evaluation of existing model capabilities, then it has a contribution but is not an ICLR-level paper.

**Questions:**

1. Details of Data Definition:
- Automated Filtering: Batches of generated instructions first undergo automated filtering.
- Multi-Round Model Validation: This is followed by a second, more rigorous validation stage involving multiple model evaluations.
- Final Retention: Only tasks that pass all validation steps are retained as the final instructions.
What are the specific details? What insights are gained?
2. What are your next steps or ideas for addressing the failure cases?

---

> ### Author Response · Authors · 2025-11-23
> **Rebuttal (1/5)**
>
> We sincerely appreciate the reviewer for highlighting the strengths of VeriWeb, noting that the `fine-grained decomposition is essential` and that `our proposed dataset effectively serves its purpose` by revealing significant performance gaps and current agent limitations. We have carefully revised the manuscript according to your valuable suggestions. Below, we address the main points raised in the review.
>
> ---
>
> **[W1.1] However, there is no particularly detailed justification for why evaluation should be conducted through subtasks (and other agent evaluation papers have proposed similar evaluation methods and metrics).**
>
> We thank the reviewer for the thoughtful comments. (1) Our goal is to benchmark long-horizon web information-seeking, where solving a task typically requires both breadth-oriented coverage and depth-oriented reasoning. Because these tasks are very challenging, **purely final-answer evaluation tends to collapse many tasks into a single zero outcome** and provides little guidance about which part of the interaction actually fails. This is why we **explicitly** decompose each task into a sequence of executable subtasks, where an agent can be **initialized from any subtask and evaluated at that level**. This enables us to measure not only overall task success but also task completion rates at different stages of the trajectory.
>
> (2) We observe that the first subtask often emphasizes breadth search, while subsequent ones focus on depth search. Therefore, we design two specific subtask-based settings: (a) Breadth-oriented search, which evaluates only the first subtask. (b) Depth-oriented search, which evaluates the overall task given the result of the first subtask. During the rebuttal period, we have additionally included more results on these settings from different agents, which further support our subtask-based evaluation protocol. A key observation in Table R1 is that, while the two settings exhibit comparable Completion Rates (CR), breadth-oriented tasks often achieve noticeably higher Success Rates (SR). The only exception is GPT-4.1, which performs poorly in breadth-oriented tasks. In breadth-oriented search, the pipeline is relatively shallow and subtasks are weakly coupled, so local retrieval errors rarely invalidate the entire instance and models can more easily produce fully correct outputs, which leads to higher SR. In depth-oriented search, later steps strictly depend on earlier ones, so errors in intermediate retrieval, reasoning, or synthesis accumulate and often cause the whole task instance to fail, sharply reducing SR. Taken together, **these findings highlight that subtask-level evaluation is essential for disentangling breadth versus depth capabilities and for precisely localizing where long-horizon web agents fail along the interaction trajectory**. *These new results have been updated in Section 4.2 (Page 8) of the revised manuscript.*
>
> Table R1 (Table 3 in the revision): Comparison of different subtask-based settings on VeriWeb.
> |Model|Breadth-Oriented||Depth-Oriented||Overall||
> |-|-:|-:|-:|-:|-:|-:|
> ||SR (%)|CR (%)|SR (%)|CR (%)|SR (%)|CR (%)|
> |**Deep Research Agent**|||||||
> |Gemini-2.5-Flash|17.5|43.1|7.6|36.1|6.9|31.2|
> |OpenAI-o4-mini|17.2|37.4|8.3|43.4|6.6|25.9|
> |**Search Engine Agents**|||||||
> |OpenAI-o3|18.2|38.7|10.6|44.3|6.0|26.1|
> |GPT-5|23.2|38.6|15.6|45.5|9.9|32.5|
> |**Browser-Use Agents**|||||||
> |Claude-3.7-Sonnet|16.6|37.8|10.6|40.1|6.0|25.0|
> |Claude-4.0-Sonnet|11.3|20.8|6.0|18.6|3.3|12.2|
> |GPT-4.1|9.6|31.5|11.9|43.4|3.6|21.4|
> |GPT-5|12.9|19.7|4.6|17.9|3.0|14.6|
>
> (3) To the best of our knowledge, VeriWeb is the first **web-based information-seeking benchmark** that **explicitly decomposes long-horizon tasks into a hierarchy of verifiable subtasks**. Existing web-agent benchmarks typically report only final task success and do not provide reusable, human-annotated sequences of intermediate goals with verifiable answers, nor do they support initializing and evaluating agents from arbitrary intermediate subtasks.

---

> ### Author Response · Authors · 2025-11-23
> **Rebuttal (2/5)**
>
> **[W1.2] Furthermore, the analysis of failure cases lacks sufficient depth and does not offer unique insights.**
>
> Thanks for your valuable suggestion. (1) We have additionally conducted a quantitative breakdown of different failure modes using GPT-5 as a judge, including (a) misinformation, (b) incomplete result, (c) retrieval failure, and (d) irrelevant result. The detailed evaluation prompt is provided in Appendix C. Note that a single response may belong to several failure modes at the same time.
>
> The results in Table R2 show that misinformation and incomplete results are the two dominant failure modes across all agent paradigms, while retrieval failures and irrelevant results occur less frequently but are still non-negligible. Deep research and search engine agents tend to fail primarily through misinformation, suggesting that they often retrieve relevant evidence but hallucinate or approximate key numerical or factual details. In contrast, the browser-use agents and multi-agent systems are more frequently dominated by incomplete results, indicating that these agents are better at locating relevant sources but struggle to aggregate and exhaustively cover all required items in long-chain tasks. Across all settings, retrieval failures rarely account for the majority of errors, highlighting that the central bottleneck lies in accurate synthesis and coverage rather than merely finding at least one relevant document. *These new results have been updated in Section 4.4 (Page 10) of the revised manuscript.*
>
> (2) Moreover, we also observe several systemic limitations. First, the agents often demonstrate shallow search behavior, invoking tools only a few times and stopping early. This limits their ability to perform comprehensive, multi-hop retrieval. Second, the agents often formulate web queries using full sentences rather than concise keywords, leading to suboptimal results and reduced accuracy.
>
> Table R2 (Table 5 in the revision): Comparison of different failure modes on VeriWeb.
> |Model|Misinformation|Incomplete Result|Retrieval Failure|Irrelevant Result|
> |-|-|-|-|-|
> |**Deep Research Agent**||||
> |Gemini-2.5-Flash|44\% (254)|34\% (197)|14\% (82)|8\% (42)|
> |Gemini-2.5-Pro|44\% (213)|36\% (172)|11\% (54)|9\% (38)|
> |OpenAI-o4-mini|50\% (249)|36\% (178)|9\% (46)|5\% (20)|
> |OpenAI-o3|63\% (248)|30\% (118)|3\% (12)|4\% (11)|
> |**Search Engine Agents**||||
> |OpenAI-o3|62\% (226)|31\% (113)|4\% (16)|3\% (9)|
> |GPT-5|46\% (172)|42\% (158)|6\% (23)|6\% (16)|
> |**Browser-Use Agents**||||
> |Qwen-VL-Max|12\% (51)|67\% (274)|14\% (58)|7\% (24)|
> |DeepSeek-V3.1|21\% (100)|55\% (262)|17\% (81)|7\% (27)|
> |Gemini-2.5-Flash|25\% (125)|53\% (258)|12\% (58)|10\% (40)|
> |Gemini-2.5-Pro|34\% (149)|49\% (214)|12\% (53)|5\% (19)|
> |Claude-3.7-Sonnet|34\% (189)|38\% (212)|19\% (105)|9\% (40)|
> |Claude-4.0-Sonnet|19\% (85)|58\% (253)|16\% (73)|7\% (24)|
> |OpenAI-o3|50\% (183)|39\% (144)|5\% (20)|6\% (16)|
> |GPT-4o|22\% (125)|47\% (269)|24\% (140)|7\% (34)|
> |GPT-4.1|33\% (216)|37\% (246)|24\% (158)|6\% (34)|
> |GPT-5|17\% (81)|55\% (254)|20\% (94)|8\% (28)|
> |**Multi-Agent Systems**||||
> |Qwen3-235B|23\% (97)|56\% (239)|12\% (53)|9\% (31)|
> |DeepSeek-V3.1|12\% (55)|61\% (270)|17\% (76)|10\% (36)|
> |OpenAI-o3|42\% (168)|41\% (163)|8\% (32)|9\% (32)|
> |GPT-5|19\% (79)|60\% (243)|13\% (56)|8\% (25)|

---

> ### Author Response · Authors · 2025-11-23
> **Rebuttal (3/5)**
>
> **[W1.3] If this paper merely defines a new benchmark data generation process (with missing details on the data synthesis process) and conducts a certain evaluation of existing model capabilities, then it has a contribution but is not an ICLR-level paper.**
>
> Thanks for your comment. (1) Regarding the missing details on the data synthesis process, we apologize for the earlier lack of clarity. We have additionally updated a more complete description of both task instruction construction and human demonstration collection in Appendix A \& B (Pages 16-23) of the revised manuscript.
>
> (2) We would also like to clarify that our work **does not** aim to propose a new data synthesis algorithm. As stated in the introduction, our main contribution is the **high-cost, human-annotated dataset** of 302 verifiable long-chain task trajectories across various real-world domains, capturing both **long-chain complexity and subtask-level verifiability**. On top of this dataset, we design the VeriWeb benchmark, which supports fine-grained analysis of existing agent capabilities.
> - While constructing challenging instructions is relatively easy for modern LLMs, obtaining correct, verifiable answers and full trajectories is the difficult part. We therefore invested substantial human effort: 50 annotators and 5 reviewers (all with at least a bachelor’s degree), with an annotation cost of \\$50 and a review cost of \\$20 per sample, for a total base budget of roughly \\$21K. We have released this dataset and its trajectories to the community, and we believe it constitutes a **highly valuable resource for research on information-seeking agents**.
> - Beyond our response to [W1.1], we would like to further emphasize why our benchmark design is central. Existing information-seeking benchmarks largely fall into two categories. The first targets single-fact retrieval, which is fully **verifiable** but fails to capture realistic tasks that require information-rich, synthesized outputs. The second targets deep research or report-style generation, which captures open-ended **uncertainty** but makes direct ground-truth evaluation difficult, and often relies on rubric-based scoring for subjective aspects. In contrast, our primary goal with VeriWeb is to benchmark one core capability that remains challenging in realistic web environments: **large-scale retrieval and synthesis of information from diverse sources (similar to deep research) that still yields a verifiable answer (similar to single-fact retrieval)**.
>
> This is also why, when submitting to ICLR, **we selected "Datasets and Benchmarks" as the primary area**. Our focus is not on methodological novelty, but on providing a challenging, carefully curated benchmark and dataset. In this context, we would like to note that ICLR regularly accepts a substantial number of benchmark-focused papers each year (e.g., searching "benchmark" on DBLP shows 125 such papers accepted at ICLR 2025). This suggests that contributions like ours, which provide new datasets and benchmarks to the community, are well aligned with the scope and standards of ICLR.

---

> ### Author Response · Authors · 2025-11-23
> **Rebuttal (4/5)**
>
> **[Q1.1] Automated Filtering: Batches of generated instructions first undergo automated filtering.**
>
> Sorry for the confusion. The automated filtering stage is a purely rule-based procedure implemented in code and does not invoke any large language model. Its goal is to remove obviously malformed or trivial outputs before any model-based validation. Concretely:
> - **Format and schema checks.** We parse the model outputs as JSON and enforce a fixed schema. Each entry must be a JSON object containing exactly the fields "instruct" and "subtasks". The "instruct" field must be a non-empty string. The "subtasks" field must be a list whose elements are subtask objects. Each subtask must contain all required fields, each with the correct type, and no additional fields are allowed. Entries that cannot be parsed as valid JSON, that contain missing or extra fields, or that have empty required fields are discarded at this stage.
> - **Simple difficulty filtering.** In addition to syntax and type checks, we apply simple structural heuristics as a coarse proxy for difficulty. In particular, tasks with too few subtasks (for example, a single-step instruction) are filtered out automatically because they are unlikely to represent the multi-step, web-oriented behavior that we target. This filtering is implemented via deterministic rules on the number of subtasks.
>
> Only instructions that satisfy all of these deterministic checks are passed to the second stage. *These clarifications have been updated in Appendix A.1 (Page 17) of the revised manuscript.*
>
> ---
>
> **[Q1.2] Multi-Round Model Validation: This is followed by a second, more rigorous validation stage involving multiple model evaluations.**
>
> The second stage is a model-based validation that operates only on the subset of instructions that have passed the automated filtering described above. We use a strong LLM (In our early experiments, we primarily used GPT-4.1. After GPT-5 was released, we switched to GPT-5.) as an automatic reviewer and instruct it via a detailed prompt to evaluate each candidate instruction with respect to several criteria, including (1) Structure and format compliance, (2) Task difficulty and long-chain nature, (3) Verifiability and uniqueness, (4) Feasibility, (5) Innovation and diversity, and (6) Reasonableness of scope limitations. The detailed validation prompt is provided in Appendix A.2.
>
> For each candidate, the review model outputs one of three labels:
> - **"Pass":** the instruction is accepted and moves to the next round or to the final pool.
> - **"Revise":** the instruction is useful in principle but has concrete, fixable issues. These cases are returned to the generation phase with feedback, revised, and then re-submitted to the review model.
> - **"Reject":** the instruction is fundamentally flawed or not worth revising and is discarded.
>
> We run 3 model-based validation rounds for each instance. The motivation is to reduce the probability that a low-quality instance passes due to a single stochastic error of the reviewer model. Only instances that are consistently labeled as "Pass" across all model-based validation rounds survive this stage. *These clarifications have been updated in Appendix A.2 (Page 17--20) of the revised manuscript.*
>
> ---
>
> **[Q1.3] Final Retention: Only tasks that pass all validation steps are retained as the final instructions. What are the specific details? What insights are gained?**
>
> (1) The final instruction set used in our experiments consists of those tasks that (i) satisfy the deterministic automated filtering in [Q1.1] and (ii) are labeled as "Pass" in all rounds of model-based validation described in [Q1.2]. Moreover, a subset of these high-quality instructions is recycled as additional seed instructions in later generation cycles, which further enriches the diversity of subsequent batches.
>
> (2) Overall, we find that careful design of seeds and generation prompts has a larger impact on final dataset quality than increasing validator complexity, and that the remaining challenges center on task diversity and subtle feasibility rather than basic grammatical correctness.
> - We observe that generation quality is more critical than validation complexity. In practice, using a strong generator model together with hand-curated seed instructions and well-designed prompts already yields high grammatical and semantic quality. By comparison, adding more validation provides smaller gains than further improving the seeds and generation prompt design.
> - We further find that the remaining difficulties lie primarily in expanding the diversity of tasks and in identifying instructions whose feasibility is hard to assess automatically. Clearly unsatisfiable or incoherent tasks are already filtered out by our current pipeline. In contrast, diversity and subtle feasibility issues are better addressed through richer seeds and refined generation prompts than through increasingly elaborate validation heuristics.

---

> ### Author Response · Authors · 2025-11-23
> **Rebuttal (5/5)**
>
> **[Q2] What are your next steps or ideas for addressing the failure cases?**
>
> We identify four recurring failure modes: (a) misinformation, (b) incomplete results, (c) retrieval failures, and (d) irrelevant results, as well as two systemic issues: shallow search and suboptimal query formulation. These observations directly motivate the following research directions:
>
> - **Memory-enhanced agents (shallow search, incomplete results).** Many errors arise because agents stop after retrieving only part of the necessary evidence. The subtask decomposition and verifiable answers in VeriWeb allow researchers to model long-horizon browsing as a sequential decision process and to investigate memory-enhanced agents that maintain explicit coverage targets or checklists across subtasks, and learn policies for when to continue exploring versus when to terminate. This connects directly to work on agentic LLMs with structured memory and long-term state tracking [1,2,3].
>
> - **Efficient query and search policies (retrieval failures, irrelevant results).** A second class of failures comes from suboptimal use of the search tool, such as overly long, unfocused queries or premature narrowing of the search space. VeriWeb exposes these decisions at a fine granularity, enabling the study of agents that learn query strategies for web search using reinforcement learning [4,5], including how to decompose a subtask into multiple targeted queries and how to control search depth. Subtask-level feedback efficiency provides natural signals for optimizing such search policies.
>
> - **Self-verifying reasoning (misinformation).** We also observe cases where the correct pages are retrieved, but the final answer is numerically wrong or not fully supported. This suggests a need for self-verifying reasoning [6,7]: agents that explicitly cross-check multiple sources, align each answer field with supporting evidence, and internally verify the consistency of their intermediate conclusions before committing to a final answer. The subtask-level verifiable answers in VeriWeb make it possible to evaluate and improve such evidence-grounded reasoning strategies.
>
> ---
>
> **References**
>
> [1] MemoryBank: Enhancing Large Language Models with Long-Term Memory. AAAI 2024.
>
> [2] A-mem: Agentic memory for llm agents. arXiv 2025.
>
> [3] Disentangling memory and reasoning ability in large language models. ACL 2025.
>
> [4] Search-r1: Training llms to reason and leverage search engines with reinforcement learning. arXiv 2025.
>
> [5] R1-Searcher++: Incentivizing the Dynamic Knowledge Acquisition of LLMs via Reinforcement Learning. arXiv 2025.
>
> [6] WebThinker: Empowering Large Reasoning Models with Deep Research Capability. arXiv 2025.
>
> [7] WebWeaver: Structuring Web-Scale Evidence with Dynamic Outlines for Open-Ended Deep Research. arXiv 2025.

---

> ### Author Response · Authors · 2025-11-27
> **Looking Forward to your Reevaluation**
>
> Dear Reviewer 76ET,
>
> We are glad that the reviewer appreciates our attempt, and sincerely thank you for the constructive comments. As suggested, we have additionally provided detailed clarifications for our subtask-based evaluation and human demonstration collection, and included further discussions and experiments on failure cases. Please let us know if you have other questions or comments.
>
> Since the discussion window has less than a week remaining, we sincerely look forward to your reevaluation of our work and would very appreciate it if you could raise your score to boost our chance of more exposure to the community. Thank you very much!
>
> Best regards,
>
> Authors of VeriWeb

---

### Meta-Review · Area_Chair_tfP8 · 2026-01-18

**Summary:**

## Concerns over the Dataset Correctioness
The first major concern posed by reviewers Q1hS is the annotation quality for complex tasks. I shared the concern and am worried about its potential impact on the research community if not thoroughly re-evaluated: the community may mistakenly consider a model that overfit errors on the benchmark to be "good". For example, from Figure 6 (error analysis), (1) consider 39M growth to be an error with the ground truth being 38.64M without specific instructions on the precision (2) the question asks for "the name of the VFX company that contributed to the most episodes", but the ground truth has multiple companies. Both error verdicts seem to be overly strict. Human evaluator comparison also indicate an over-strictness in the evaluation.

As a benchmark paper, the correctness is the most important and basic requirement. From the content provided in the paper, it is unclear about the quality of the benchmark: annotation ambiguity as above could just be corner cases, or could be prevalent in the entire dataset. With this level of uncertainty, I can only recommend a rejection without further information available.

## Other concerns
Some unaddressed concerns that is not as critical as above but also informed my decision.
### Over-simplification
Reviewer Vcuj poses a concern about underrepresenting real world scenarios when only subtask-verifiable tasks are considered. I agree to this concern. Moreover, by presenting the chain of subtasks to the LLM agents, a clear roadmap to deeper multi-hop questions is clearly revealed in the task, which also reduces the complexity of real world information seeking tasks where only the final request may be presented.
### Existence of Other Benchmarks with Subtask Metrics
Reviewer tn1J is in particular concerned about AgentCompany. Although I agree with authors that the task types and several aspects are different, the core idea of subtask evaluation is similar. This concern does not consitute as a strong rejection signal, but does reduce the novelty of this work.

**Reviewer Concerns:**

Outstanding concerns are presented in the summary.

The reviewers' concerns over the quality of LLM-as-a-judge is properly responded. It is not a perfect alignment, but definitely with a reasonable correlation. So it should be considered as mostly addressed.

Another shared concersn among several reviewers is the lack of human baselines. The authors does provide a human baselines with limited time, which is no better than best-performing models.

The authors also provided more details on the benchmark creation process which was initially missing in the manuscript and questioned by several reviewers.

**Reviewer Scores:**

I don't think reviewer are changing their scores. The authors claimed positive feedback from reviewer tn1J, but I think the reviewer still holds concern over the AgentCompany comparison, and were unlikely to change scores even if the rebuttal window had not closed.

---

### Decision · Program_Chairs · 2026-01-26

Reject